# Producers Equilibria and Dynamics in Engagement-Driven Recommender Systems

**Krishna Acharya**                                    *krishna.acharya@gatech.edu*

*Georgia Institute of Technology*

**Varun Vangala**                                       *vvangala3@gatech.edu*
*Georgia Institute of Technology*

**Jingyan Wang**                                        *jingyanw@ttic.edu*
*Toyota Technological Institute at Chicago*

**Juba Ziani**                                          *jziani3@gatech.edu*
*Georgia Institute of Technology*

**Reviewed on OpenReview:** *https://openreview.net/forum?id=EWT4GxjGDS*

## Abstract

Online platforms such as YouTube, Instagram heavily rely on recommender systems to decide what content to present to users. Producers, in turn, often create content that is likely to be recommended to users and have users engage with it. To do so, producers try to align their content with the preferences of their targeted user base. In this work, we explore the equilibrium behavior of producers who are interested in maximizing user *engagement*. We study two variants of the content-serving rule for the platform's recommender system, and provide a structural characterization of producer behavior at equilibrium: namely, each producer chooses to focus on a single embedded feature. We further show that specialization, defined as different producers optimizing for distinct types of content, naturally emerges from the competition among producers trying to maximize user engagement. We provide a heuristic for computing equilibria of our engagement game, and evaluate it experimentally. We highlight i) the performance and convergence of our heuristic, ii) the degree of producer specialization, and iii) the impact of the content-serving rule on producer and user utilities at equilibrium and provide guidance on how to set the content-serving rule [1].

## 1 Introduction

Recommender systems have transformed our interactions with online platforms like Instagram, Spotify, and YouTube (Stigler Committee, 2019; Qian and Jain, 2022). These systems curate content to enhance user experiences, fostering user retention and engagement (Goodrow, 2021; Instagram, 2023). This often translates into increased revenue for the platform. Due to their importance to today's digital industry, there has been a large body of work aiming at developing new and improving existing recommendation algorithms, e.g., (Koren et al., 2009; Li et al., 2010; Lü et al., 2012; Wang and Zhang, 2012; Luo et al., 2014; Covington et al., 2016; He et al., 2017; Yi et al., 2019).

In 2023, the creator economy, driven largely by these systems, was valued at a staggering \$250 billion (Perelli, 2023). Content producers have adapted to this landscape to act *strategically* (Milli et al., 2023; Merrill and Oremus, 2021; Mack, 2019). They often compete against each other and tailor their content to maximize *exposure* (or how many users does a producer reach) or *engagement* (or how much users engage with a given

---

[1]Code available at `https://github.com/krishnacharya/recsys_eq`

producer's content). This competition can be modelled as a game, where understanding the dynamics offers insights into content creation incentives and phenomena like content specialization. However, comprehending these dynamics, both theoretically and empirically, presents challenges when considering the complex nature of user preferences and content strategies in high-dimensional spaces. Finding Nash equilibria is in general a computationally difficult process, especially as the number of players and the action space increase. In the case of recommender systems, user preferences and producer contents are represented by high-dimensional vectors, and the set of possible strategy profiles for the producers rapidly grows intractably large as the size of the game increases.

In this work, we aim to provide new insights into producer competition and the equilibria of recommender systems. We take a departure from much of the related literature that aims at understanding equilibria and dynamics when producers try to maximize *exposure* (i.e., how many users see their content). We instead focus on producers interested in maximizing *engagement*, a metric that encompasses not just exposure but also how much users interact with content.

Our main goal is to understand the extent and conditions under which content specialization occurs—defined as different producers choosing to create distinct types of content—instead of all producers producing the same homogeneous content.

**Summary of contributions**  Our main contributions are as follows:

- In Section 2, we formally introduce our model. We assume producers aim to maximize user *engagement* instead of user *exposure*, where the latter is typical in works characterizing producer equilibria in recommender systems (Hron et al., 2022; Jagadeesan et al., 2022). We rely on the softmax rule used in previous work (Hron et al., 2022; Chen et al., 2019) for showing content to users, but also introduce a new linear-proportional content-serving rule as a baseline for comparison.

- In Section 3, we provide our main theoretical structural characterization result. We *mathematically prove* that at equilibrium, producers prefer producing content that targets a single (embedded) feature at a time, rather than investing across several features.

- In Section 4, we characterize the pure Nash equilibrium with the linear content-serving rule for a simplified setting in which we assume all users are "single-minded", i.e. are only interested in a single type of content. The closed-form equilibrium we derive exhibits *specialization*, defined as when different producers split themselves across different types of content, rather than all producing the same, homogeneous content. While this setting is simplified, we note in the experimental results of Section 5 that the insights it provides do carry on to the general softmax content-serving rule and to more general user preferences.

- In Section 5, we present experiments on synthetic data and three real-world datasets—Movielens-100k, Amazon Music, and Rent the Runway (Harper and Konstan, 2015; Ni et al., 2019; Misra et al., 2018). First, we introduce a computationally efficient heuristic based on best-response dynamics (Algorithm 1) for computing pure Nash equilibria of our engagement game. We observe that this heuristic almost always converges to a pure Nash equilibrium within relatively few steps. Second, our experiments further characterize producer specialization, showing that it occurs even under more complex settings than those in Section 4. We then study the effect of the temperature parameter in the softmax content-serving rule on both producer specialization and utility. Specifically, we demonstrate that the degree of producer specialization is monotonic with respect to temperature: higher temperatures (representing greater exploration in the content shown to users) incentivize more homogeneous content production, while lower temperatures (favoring the most relevant content for each user) encourages specialization, with producers focusing on distinct types of content. Lastly, we show that both producer and user utilities decrease monotonically with increasing temperature. A low softmax temperature yields the highest utility for both producers and users, emphasizing the benefits of selecting lower temperatures in the softmax content-serving rule.

## 1.1 Related work

The study of strategic behavior and incentives among producers in recommender systems has seen much interest recently. These interactions can broadly be classified into two types of games: exposure-based, where producers are rewarded for maximizing the reach of their content, and engagement-based, where producer rewards depend not only on reach but also on how well the recommended content aligns with user preferences.

**Exposure games** The seminal works of Raifer et al. (2017); Basat et al. (2017); Ben-Porat and Tennenholtz (2018); Ben-Porat et al. (2019b;a; 2020) introduce game-theoretic models of competition among producers aiming to maximize exposure. While these early studies constrain content creators to finite strategy spaces by requiring them to select from a pre-specified, finite catalog, more recent works—such as Jagadeesan et al. (2022); Hron et al. (2022), which are most closely related to ours—have relaxed this assumption by focusing on creators with continuous strategy sets.

**Engagement games** The works of (Yao et al., 2023a; Immorlica et al., 2024; Huttenlocher et al., 2024) model producer rewards based on engagement. Yao et al. (2023a) study social welfare in scenarios where content creators compete without the mediation of a central recommender, and focus on bounding the Price of Anarchy[2]. Immorlica et al. (2024) examine engagement games from a different perspective, exploring the trade-off between producing high-quality content and gaming the recommender system by creating low-quality, "clickbait" content. Huttenlocher et al. (2024) model the problem as a two-sided marketplace with departing users and creators and show that maximizing total engagement in such a setting is NP-hard.

**Mechanism design** Works on Exposure and Engagement games focus on characterizing producer competition and equilibrium under a fixed content-serving rule and producer reward. In contrast, Yao et al. (2023b; 2024) adopt a mechanism design perspective, designing rewards and serving rules incentivizing equilibria that have high social welfare. Hu et al. (2023) explore a similar setting but model the platform as a linear contextual bandit.

**User dynamics** Our work studies adaptive producers who change their content vectors to maximize user engagement. The user preferences are static, consistent with the literature on Exposure and Engagement games. Another line of work by (Dean and Morgenstern, 2022; Kalimeris et al., 2021) studies shifting user preferences based on the content recommended to them. Lin et al. (2024) adopt this user preference shift model but also model evolving producers, providing conditions for user polarization, though their producer evolution is not game-theoretic.

A table providing a more detailed comparison of our paper to related work along various axes (such as the nature of producer reward, type of equilibrium and dynamics) is available in Appendix A.

## 2 Our model

We consider an *engagement game* between $n$ producers on an online platform. The producers must decide what type of content to produce in a $d$-dimensional space to maximize the engagement from users. This space is generally not what one may think of as the original feature space, where features are defined, e.g., as different genres that a producer or user may care about. Rather, these features are embedded features that are the results of a matrix factorization algorithm[3] whose goal is to learn representative "directions" of the recommendation problem, as in Hron et al. (2022); Jagadeesan et al. (2022). The online platform then uses their recommender system to recommend content to users as a function of how well producer content matches user preferences. More formally:

**Producer model** We have $n$ producers on the platform. Each producer $i$ *chooses* a content vector $s_i$ from the set $\mathcal{S} := \{s : s \in \mathbb{R}^d_{\geq 0}, \|s\|_1 \leq 1\}$. We note that we focus on the $\ell_1$-norm in order to model the relative

---

[2]The ratio between the optimal welfare to that at the decentralized equilibrium(Koutsoupias and Papadimitriou, 1999)
[3]MF learns latent representations of users and movies which are then used to predict user preferences and ratings. Our results however do not depend on the specifics of the algorithm used to obtain this embedding.

amount of weight that each producer has on each embedded feature. We let $\Delta\mathcal{S}$ be the set of probability distributions over $\mathcal{S}$. Letting $s_i(f)$ denote the $f$-th entry of content vector $s_i$ for feature $f \in [d]$, one can interpret $s_i(f)$ as the fraction of producer $i$'s content that targets embedded feature $f$.

**User model**   We have $K$ users on the platform. Each user $k \in [K]$ is described by a preference vector in $\mathcal{C} := \{c : c \in \mathbb{R}^d_{\geq 0}, \|c\|_1 \leq 1\}$. We note that $c_k$ describes user preferences in the form of the amount of weight they attribute to each embedded feature. The more weight on the feature, the more utility the user derives from seeing content that aligns with said feature. We assume that the utility of a user with preferences $c_k$ who faces content $s$ is given by $c_k^\top s$.

**Recommender system's content-serving rule**   The platform uses a recommender system (RS) to decide which producer's content to show to which user. Denote $\vec{s} = (s_1, \ldots, s_n) \in \mathcal{S}^n$ the full profile of producers' production choices. The RS shows producer $i$'s content $s_i$ to a user with preferences $c_k$ with probability $p_i(c_k, \vec{s}) = p_i(c_k, s_i, s_{-i})$, where $s_{-i}$ denotes the rest of the producers. We call this probability the *content-serving rule*. In this paper, we consider two content-serving rules:

1. The *softmax* content-serving rule is, as in  Hron et al. (2022); Chen et al. (2019):

$$p_i(c, s_i, s_{-i}) \triangleq \frac{\exp\left(\frac{c^\top s_i}{\tau}\right)}{\sum_{j=1}^n \exp\left(\frac{c^\top s_j}{\tau}\right)} \tag{1}$$

   where $\tau$ denotes the softmax temperature. A low temperature corresponds to greedier serving (i.e., only the best fitting producer's content is shown to the user), whereas a high temperature corresponds to adding more randomness to the serving ("worse" producers may still have their content shown to the user, albeit with lower probability). The limit $\tau \to 0$ corresponds to a hard maximum, i.e., the producer whose content is best aligned, namely producer $\arg\max_{j \in [n]} c^\top s_j$, is shown to user $c$.

2. The *top-k softmax* content-serving rule first selects the top-$k$ producers with the highest alignment scores $c^\top s_j$, where $j \in [n]$. Among the selected $k$ producers, we then apply the softmax function with temperature $\tau$:

$$p_i(c, s_i, s_{-i}) \triangleq \begin{cases} \dfrac{\exp\left(\frac{c^\top s_i}{\tau}\right)}{\sum_{j \in \mathcal{K}} \exp\left(\frac{c^\top s_j}{\tau}\right)} & \text{if } i \in \mathcal{K}, \\ 0 & \text{otherwise,} \end{cases} \tag{2}$$

   where $\mathcal{K} = \{j \in [n] : c^\top s_j \text{ is among the top } k \text{ values of } \{c^\top s_1, \ldots, c^\top s_n\}\}$. When $k = n$, the top-$k$ softmax rule reduces to the regular softmax rule defined in (1), and when $k = 1$, it's defined as the *greedy serving rule*.

3. The *linear-proportional* content-serving rule (Luce (1977) choice axiom), where each producer's content $s_i$ is shown to user $c_k$ with a probability directly proportional to $c_k^\top s_i$.

$$p_i(c, s_i, s_{-i}) \triangleq \begin{cases} \dfrac{c^\top s_i}{\sum_{j=1}^n c^\top s_j} & \text{if } c^\top s_i > 0, \\ 0 & \text{if } c^\top s_i = 0. \end{cases} \tag{3}$$

   Note that the linear serving rule is well-defined even when $c^\top s_j$ is zero for all producers $j$. We use this rule as a baseline to compare performance to the typical softmax-based rule, and as an alternative rule to derive theoretical insights (our experiments in Section 5 show that theoretical insights for the linear-content serving rule in fact extend to the softmax rule).

4. The *round-robin serving rule* serves producers in a cyclic order: In the first round, all users are shown producer 1's content, in the second round producer 2 and so on. Formally, the serving probability for

user $c$ in serving round $r$ is defined as:

$$p_i^r(c, s_i, s_{-i}) \triangleq \begin{cases} 1 & \text{if } i = (r-1 \mod n) + 1, \\ 0 & \text{otherwise.} \end{cases} \tag{4}$$

**Producer utilities** One of our contributions is to characterize the utility of a producer using *engagement*, rather than just *exposure*. In exposure games, the utility of a producer is simply the probability that this producer's content is shown to a user. In contrast, with *engagement*, the utilities incorporate an additional term that measures *how much a user engages with the content once this content is shown to them.* Formally, we assume that a producer $i$ who successfully shows content $s_i$ to user $k$ with preferences $c_k$ derives utility $s_i^\top c_k$ from that user. This captures the fact that a user whose preferences are better aligned with the producer's content are more likely to spend more time engaging with that content. Formally, we define the *expected engagement utility* for producer $i$ as

$$u_i(s) = u_i(s_i, s_{-i}) \triangleq \sum_{k=1}^{K} p_i(c_k, \vec{s}) \cdot c_k^\top s_i. \tag{5}$$

Note that this *expected* utility is reweighted by the probability of producer $i$ showing $s_i$ to user $k$, as $i$ derives no utility from user $k$ if said user does not see his content in the first place. The total producer utility $U_p$ is then defined as

$$U_p \triangleq \sum_{i=1}^{n} \sum_{k=1}^{K} p_i(c_k, \vec{s}) \cdot c_k^\top s_i.$$

**User utilities** We similarly define the utility for a user with embedding $c_k$ as its engagement in expectation across all producers i.e., $\sum_{i=1}^{n} p_i(c_k, \vec{s}) \cdot c_k^\top s_i$. The total user utility $U_u$ is defined as

$$U_u \triangleq \sum_{k=1}^{K} \sum_{i=1}^{n} p_i(c_k, \vec{s}) \cdot c_k^\top s_i.$$

**Remark 2.1.** *The value of the total producer utility $U_p$ and the total user utility $U_u$ are equal, and the average producer utility and the average user utility are equal up to a multiplicative factor.*

**Remark 2.2.** *For the round-robin serving rule* (4)*, it is easy to see that the utility for a producer $i$ is zero if it is not being served. When it is served, the utility is given by $\max_{s_i \in S} \sum_{k=1}^{K} c_k^\top s_i = \left\| \sum_{k=1}^{K} c_k \right\|_\infty$, which is achieved by setting $s_i$ to the basis vector corresponding to the largest weight.*

## 3 Equilibrium Structure

In this section, we derive our main structural result for equilibrium in engagement games: namely, we show that at equilibrium, *each producer prefers targeting a single embedded feature at a time.*

Our first main assumption is that for all users, their features are strictly positive:

**Assumption 3.1.** *For every user $k$ and feature $f$, we have $c_k(f) > 0$.*

This assumption holds in practice when user representations are obtained via *Non-Negative Matrix Factorization* (NMF) (Lee and Seung, 2000; Luo et al., 2014) as is observed in Jagadeesan et al. (2022); Hron et al. (2022) and in our experiments in Section 5. We additionally make an assumption on the data distribution, guaranteeing that user preferences $c_1, \ldots, c_k$ are non-trivial:

**Assumption 3.2.** *For all pairs of production strategies $s$, $s' \in \mathcal{S}$ such that $s \neq s'$, there must exist at least one user $k$ such that $c_k^\top s \neq c_k^\top s'$.*

This is a mild assumption that states that there is enough diversity in user preferences. Equivalently, this assumption states that user preferences are non-trivial and diverse enough such that $span(c_1, \ldots, c_K) = \mathbb{R}^d$, i.e., the user preferences span [4] all of $\mathbb{R}^d$. If the user preferences do not span all of $\mathbb{R}^d$, this means that some

---

[4]Indeed, there then exists a subset of size $d$ of $(c_1, \ldots, c_K)$ that forms a basis for $\mathbb{R}^d$. If $s \neq s'$, they must differ in at least one coordinate in this basis, so there must exist $c_k$ such that $c_k^\top s \neq c_k^\top s'$

latent features are redundant. In practice, one may work with a reduced embedding dimension and perform a new matrix factorization until Assumption 3.2 holds.

We are now interested in understanding properties of the Nash equilibria (NE) of our engagement-based content production game, defined as the game where each producer decides which content to produce to maximize their utility. Nash equilibria are a classical concept for solving games (Nash, 1951). We apply the standard definition to our formulation as follows.

**Definition 3.3** (Nash Equilibrium)**.** *For any producer $i$, the strategy $s_i^* \in \mathcal{S}$ that solves $u_i(s_i^*, s_{-i}) = \max_{s_i \in \mathcal{S}} u_i(s_i, s_{-i})$ is called a* best response *to $s_{-i}$. A strategy profile $(s_1^*, \ldots, s_n^*) \in \mathcal{S}^n$ is a pure-strategy Nash equilibrium (pure NE) if and only if for every producer $i \in [n]$,*

$$u_i(s_i^*, s_{-i}^*) = \max_{s_i \in \mathcal{S}} u_i(s_i, s_{-i}^*).$$

*A strategy profile $(D_1^*, \ldots, D_n^*) \in \Delta \mathcal{S}^n$, where $\Delta \mathcal{S}^n$ denotes the probability simplex over $\mathcal{S}^n$, is a mixed-strategy Nash equilibrium (mixed NE) if and only if for every producer $i \in [n]$,*

$$\mathbb{E}_{s_i \sim D_i^*, s_{-i} \sim D_{-i}^*} [u_i(s_i, s_{-i})] = \max_{D_i \in \Delta \mathcal{S}} \mathbb{E}_{s_i \sim D_i, s_{-i} \sim D_{-i}^*} [u_i(s_i, s_{-i})].$$

Under either pure or mixed Nash equilibrium, all producers best respond to each other and do not want to change their strategy: i.e., each producer maximizes its utility and gets the best utility it can by playing the Nash, assuming all remaining producers also play the Nash.

We now characterize the equilibria of our game, under both content serving rules of Equation (1) and Equation (3), showing that producers prefer to focus on a single embedding dimension at a time:

**Theorem 3.4.** *Suppose Assumptions 3.1 and 3.2 hold. Let $\mathcal{B} := (e_1, \ldots, e_d)$ be the standard basis for $\mathbb{R}^d$, where each $e_j$ is the unit vector with value 1 in coordinate $j \in [d]$ and 0 in all other coordinates. Under both types of content-serving rules, if there exists a NE, any pure strategy for producer $i$ must satisfy $s_i^* \in \mathcal{B}$, and any mixed strategy must be a distribution supported on $\mathcal{B}$ in this equilibrium.*

Informally, the theorem shows that at equilibrium, producer strategies are supported on the standard unit basis rather than on the entire simplex $\mathcal{S} = \left\{ s : \|s\|_1 \le 1, \ s \in \mathbb{R}_{\ge 0}^d \right\}$. Each producer focuses on a single feature in the embedded space. Conceptually, the producers play a feature selection game where they trade-off i) choosing the "best" features (those that align best with users' preferences) to improve the reward when showing content to users with ii) choosing potentially sub-optimal features to avoid competition with other producers over top features and improve the chance of showing content to users.

The remainder of the paper studies how the choice of content-serving rule affects the trade-off between effects i) and ii). i) pushes for more homogeneous content production while ii) promotes specialization. Section 4 provides some theoretical insights on this trade-off, and Section 5 provides experimental results for both the linear-proportional and the softmax serving rule.

## 3.1 Proof of Theorem 3.4

**Preliminary properties of producers' utilities**  We start by noting the convexity properties of producers' utilities. Everywhere in this proof, we denote $S(c, s_{-i}) = \sum_{j \ne i}^n c_k^\top s_j$.

**Claim 3.5** (Convexity for *linear-proportional* serving rule)**.** *The function*

$$f_{k, s_{-i}}(x) = \frac{x^2}{x + \sum_{j \ne i}^n c_k^\top s_j}$$

*is convex in $x$ on domain the $\mathbb{R}_{>0}$. Further, suppose $\sum_{j \ne i}^n c_k^\top s_j > 0$. Then it is strictly convex in $x$ on the domain $\mathbb{R}_{>0}$.*

*Proof.* The first-order derivative is given by $f_{k, s_{-i}}'(x) = \frac{x(2S(c, s_{-i}) + x)}{(S(c, s_{-i}) + x)^2} \ge 0$. The second order derivative of $f$ is given by $f_{k, s_{-i}}''(x) = \frac{2S(c, s_{-i})^2}{(x + S(c, s_{-i}))^3}$. Note that on $x \in \mathbb{R}_{>0}$, $f_{k, s_{-i}}''(x) \ge 0$, with $f_{k, s_{-i}}''(x) > 0$ when $S(c, s_{-i}) > 0$. This concludes the proof. $\square$

**Claim 3.6** (Convexity for *softmax* serving rule). *The function*

$$f_{k,s_{-i}}(x) = \frac{x \exp(x/\tau)}{\exp(x/\tau) + \sum_{j\neq i}^{n} \exp(c_k^\top s_j/\tau)}$$

*is strictly convex in $x$ for $|x| \leq \tau \log(S_{exp})$.*

In practice, we note that we expect this condition on $x$ to hold when the game is large enough. Indeed, it is equivalent to $S_{exp} > \exp(x/\tau)$. When the number of producers grows, $S_{exp}$ also grows while $\exp(x/\tau)$ remains bounded by $\exp(1/\tau)$ (we restrict attention in this entire proof to $x$ representing an inner product of the form $c^\top s$, which we know is in $[0, 1]$).

*Proof.* Let us overload the $S$ notation and write $S_{\exp} = \sum_{j\neq i} \exp(c_k^\top s_j/\tau)$. The first-order derivative is given by

$$f'_{k,s_{-i}}(x) = \frac{(1 + x/\tau)\exp(x/\tau) S_{\exp} + \exp(2x/\tau)}{(\exp(x/\tau) + S_{\exp})^2}$$

The second order derivative is then given by

$$f''_{k,s_{-i}}(x) = \frac{S_{exp}\exp(x/\tau)}{\tau^2(S_{exp} + \exp(x/\tau))^3} \cdot (2S_{exp}\tau + S_{exp}x + 2\tau\exp(x/\tau) - x\exp(x/\tau)))$$

$$= \frac{S_{exp}\exp(x/\tau)}{\tau^2(S_{exp} + \exp(x/\tau))^2} \cdot \left(2\tau + x\frac{S_{exp} - \exp(x/\tau)}{S_{exp} + \exp(x/\tau)}\right)$$

Notice that this is strictly positive so long as $S_{exp} > \exp(x/\tau)$, concluding the proof. $\square$

Now, we note that for every producer $j$, in each strategy $s_j$ supported in a mixed strategy profile at equilibrium, we must have $s_j > 0$. Indeed, if any producer sets $s_j = 0$ and produce no content, they get a utility of 0; however, setting $s_j > 0$ leads to $s_j^\top c_k > 0$ for all users $k$, by Assumption 3.1, and yields strictly positive utility. For a mixed strategy profile to constitute an equilibrium, every action on the support of each player's strategy must have the same utility, and this utility then has to be strictly positive, so it must be that $s_j > 0$ on the entire mixed strategy's support. Then, $S(c, s_{-i}) > 0$ on any strategy profile on the support of a mixed Nash equilibrium. Therefore, we can restrict attention to $\mathcal{S}' = \mathcal{S}/\{0\}$. We know that $f_{k,s_{-i}}(x)$ is then strictly convex in $x$. Now, let's examine the function $g_{k,s_{-i}}(s_i) = f_{k,s_{-i}}(c_k^\top s_i)$. Clearly, $g_{k,s_{-i}}$ is convex and for all $k$,

$$g_{k,s_{-i}}(\lambda s_i + (1-\lambda)s_i') \leq \lambda g_{k,s_{-i}}(s_i) + (1-\lambda)g_{k,s_{-i}}(s_i').$$

Further, pick any $\lambda \in (0, 1)$ and $s_i \neq s_i'$. There must exist $k$, by Assumption 3.2, such that $c_k^\top s_i \neq c_k^\top s_i'$. Then, we have that, for that $k$,

$$g_{k,s_{-i}}(\lambda s_i + (1-\lambda)s_i') = f_{k,s_{-i}}(\lambda c_k^\top s_i + (1-\lambda)c_k^\top s_i')$$
$$< \lambda f_{k,s_{-i}}(c_k^\top s_i) + (1-\lambda)f_{k,s_{-i}}(c_k^\top s_i')$$
$$= \lambda g_{k,s_{-i}}(s_i) + (1-\lambda)g_{k,s_{-i}}(s_i'),$$

where the inequality follows by strict convexity of $f_{k,s_{-i}}(x)$. Summing over all $k$'s, we then get that

$$\sum_{k=1}^{K} g_{k,s_{-i}}(\lambda s_i + (1-\lambda)s_i') < \lambda \sum_{k=1}^{K} g_{k,s_{-i}}(s_i) + (1-\lambda)\sum_{k=1}^{K} g_{k,s_{-i}}(s_i')$$

as at least one of the inequality in the sum has to be strict. This shows that while each $g_{k,s_{-i}}$ is not necessarily strictly convex, $\sum_k g_{k,s_{-i}}$ is. Now, note that both in the linear case as per Equation 3 and in the softmax case as per Equation 1, we have that producer $i$'s utility for playing $s_i$, under a mixed strategy profile $s_{-i} \sim \mathcal{D}$ for the remaining agents, is given by

$$u_i(s_i, \mathcal{D}) = \mathbb{E}_{s_{-i} \sim \mathcal{D}}\left[\sum_{k=1}^{K} g_{k,s_{-i}}(s_i)\right]$$

Then, $u_i(s_i, \mathcal{D})$ is also strictly convex in $s_i$ (this follows immediately by writing the definition of strict convexity and by linearity of the expectation). Finally, now, suppose by contradiction that producer $i$'s best response is not a unit vector $e_f$. Then there exists $f$ with $0 < s_i(f) < 1$. In this case, note that

$$u_i(s_i, \mathcal{D}) = u_i\left(\sum_f s_i(f)e_f, \mathcal{D}\right) < \sum_f s_i(f)u_i(e_f, \mathcal{D}),$$

by strict convexity of $s_i \to u_i(s_i, \mathcal{D})$. This is only possible if there exists $f'$ such that $u_i(s_i, \mathcal{D}) < u(e_{f'}, \mathcal{D})$. This concludes the proof.

Below, we provide our heuristic for computing Nash equilibria, under Algorithm 1. Our heuristic relies on the structural result of Theorem 3.4: if a pure equilibrium exists, then it must be supported on the standard basis $\mathcal{B}$, which simplifies the best-response computation.

---

**Algorithm 1** Best Response Dynamics for Pure Equilibrium Computation

---

**Inputs**: User embeddings $(c_1, \ldots, c_K)$. Utility $u_i(s_i, s_{-i})$ for producer $i$. Max iterations $N_{max}$.
**Output**: Pure Nash equilibrium of the engagement game.

    Initialize termination variable $fin = 0$ and iteration variable $iter = 0$. Initialize producer $i$'s strategy $s_i$ uniformly at random in $\mathcal{B} = (e_1, \ldots, e_d)$.
    **while** $fin = 0$ and $iter < N_{max}$ **do**
        Produce a random permutation vector of producers 1 to $n$.
        **for** each producer $i$ in the above permutation **do**
            Set $fin = 1$;
            Compute $s_i^* = \arg\max_{s_i \in \mathcal{B}} u_i(s_i, s_{-i})$;
            **if** $u_i(s_i^*, s_{-i}) > u_i(s_i, s_{-i})$ **then**
                $s_i = s_i^*$;
                Set $fin = 0$ and exit the for loop.
            **end**
        **end for**
    **end while**
    **return** $(s_1, \ldots, s_n)$ if $fin = 1$, and $\perp$ if $fin = 0$.

---

## 4 Equilibria under Simplified User Behavior

To provide theoretical insights about engagement equilibria, we first consider a simple, special case of our framework where each user is single-minded, and is only interested in a single type of content. We also focus on the linear-proportional content serving rule of Equation (3) for tractability.

**Assumption 4.1** (Single-minded users)**.** *For any user $k$, $c_k = e_f$ for some $f \in [d]$, where $(e_1, \ldots, e_d)$ is the standard basis.*

Note that *we only make this assumption in the current section*, and that this is an assumption on *user and not producer behavior*. This simplified assumption and setting allow us to derive our first insights towards characterizing the equilibria of our recommender system, and how producers decide what type of content to produce as a function of the total user weight on each feature. Our experiments in Section 5 show that the insights we derive in this simple single-minded user setting hold for general types of users under the right choice of content-serving rule.

Under this assumption, we provide a simplified characterization of the equilibria of our game. We consider equilibria supported on the standard basis $\mathcal{B}$, following the insights of Theorem 3.4, and let $(m_1, \ldots, m_d)$ be the number of users interested in content type $e_1, \ldots, e_d$, respectively. We assume $m_f > 0$ without loss of generality; otherwise, feature $f$ brings utility to no user nor producer, and can be removed. On the procedure side, we use the notation $(n_1, \ldots, n_d)$ to denote an aggregate strategy profile where for all $f \in [d]$, a number $n_f$ of producers pick action $e_f$.

**Lemma 4.2.** *Suppose Assumption 4.1 holds. For all $f \in [d]$, let $m_f > 0$ be the number of users with $c = e_f$. Under the linear-proportional content serving rule of Equation* (3), *there exists a pure NE supported on $\mathcal{B}$ and given by $(n_1, \ldots, n_d)$ if and only if*

$$\frac{n_f}{m_f} \leq \frac{n_{f'} + 1}{m_{f'}} \qquad \text{for all } f, f' \in [d]. \tag{6}$$

The full proof is given in Appendix B.1.

**Interpretation of the equilibrium conditions** Let $\delta_f \triangleq \frac{m_f}{\sum_{f'=1}^{d} m_{f'}}$ be the fraction of users that are interested in feature $f$. Consider strategy profile $n_f = \delta_f n$, and assume $n_f$ is integer[5]. We show that this pure strategy profile is an equilibrium of our engagement game, by verifying that Condition (6) in Lemma 4.2 holds for this construction. To see this, note that the equilibrium condition is equivalent to $\frac{n_f}{\delta_f} \leq \frac{n_{f'} + 1}{\delta'_f}$. On the left-hand side, we have $\frac{n_f}{\delta_f} = n$. On the right-hand side, we have $\frac{n_{f'} + 1}{\delta_{f'}} = \frac{\delta_{f'} n + 1}{\delta_{f'}} \geq n$, and the inequality always holds.

In this equilibrium, the number of producers that pick feature $f$ is (ignoring rounding) proportional to the number of users interested in feature $f$. This aligns with intuition: at equilibrium, the probability with which each producer is recommended to a user should be the same across all features $f$; otherwise, a producer that deviates to a feature $f'$ with higher probability will be shown more often and gain more utility. We note that Hron et al. (2022) made a related observation in a different setting[6]: they note that producer content aligns with the average user weight on each embedded feature, with the average producer content being $\bar{c} = \frac{1}{K} \sum_{k \in K} c_k$ at an approximate equilibrium. However, we note that the insight of Hron et al. (2022) remains different, in that all producers produce the same homogeneous content aligned with $\bar{c}$; we show specialization and heterogeneous content production, which we believe are commonplace in practice.

## 5 Experiments

We now provide experiments that expand our understanding of producer equilibria and utilities at equilibrium for general user incentives.

### 5.1 Experimental setup

**Synthetic Data** We generate three types of user distributions for our synthetic data. In the *uniform distribution*, we generate user embeddings $c$ uniformly at random over the probability simplex. Hence, each feature is equally represented in the data in expectation. In the *skewed distribution*, we first randomly sample positive weights $w_1 \leq \ldots \leq w_d$ (sampled from the probability simplex and then sorted). We then generate a uniformly distributed user embedding, but re-weight each feature $f$ by $w_f$ to obtain $c$. This creates a non-symmetric, skewed distribution where the total user weight for each feature is proportional to $w_f$, leading to differences across features. We also generate a *sparse distribution* for which we generate user embeddings $c \in \mathbb{R}^d$ uniformly at random over the probability simplex and then apply an element-wise masking operation. This operation uses random boolean vectors $\in \{0, 1\}^d$, with 90% of its values being zero. As per our modeling assumptions, we normalize user features $c$ to have $\ell_1$-norm 1. All synthetic experiments use $K = 10,000$ users. We vary the dimension $d$ and the number of producers $n$.

**Real Data** Following prior work on producer-side competition, we use the NMF implementation in the `scikit-surprise` package (Hug, 2020) to obtain user embeddings for the MovieLens-100k dataset (Harper and Konstan, 2015); this dataset contains $100k$ ratings, 943 users and 1682 movies. We also run experiments with two other, larger-scale ratings datasets: AmazonMusic rating (Ni et al., 2019) and RentTheRunway clothing rating (Misra et al., 2018), These datasets have around $840k$ and $100k$ unique users respectively.

---

[5]When $\delta_f n$ is not integral, we can instead round carefully so that $n_f$'s still add to $n$, and the proposed strategy remains an approximate NE, in that the equilibrium conditions are satisfied up to a small additive slack factor

[6]They use exposure instead of engagement, different assumptions on user preferences, and the softmax content-serving rule

The insights from our experiments on these datasets turn out to be similar to the Movielens-100k dataset and are deferred to Appendix F.

**Equilibrium computation: a best-response based heuristic**   We provide a simple heuristic based on best-response dynamics to compute pure-strategy Nash equilibria in Algorithm 1. In each iteration, it goes through the list of producers in randomly sorted order. For each producer, it computes the best basis vector response of the producer. If the producer is already playing a best response, the algorithm goes to the next producer; otherwise, the algorithm notes the producer is not currently best responding, updates the producer's strategy to his best response, and starts the next iteration. If at the end of the for loop, we realize that all producers are playing a best response, we have found a NE, hence we stop and output the current strategy profile. If after $N_{max}$ iterations, we still have not found a NE, we output $\perp$ to signal that our dynamics did not converge.

We will see that our heuristic terminates the large majority of the time in our experiments; in that case, it must output a pure Nash Equilibrium, as the algorithm can only terminate when all producers best respond to each other.[7]

## 5.2   Experimental results

**Convergence of Algorithm 1 and rate of convergence**   For our engagement game, we consider 12 different numbers of producers $(2, 5, 10, 20, \ldots, 100)$ and 6 different embedding dimensions $(5, 10, 15, 20, 50, 100)$. Each of these $12 \times 6$ configurations is instantiated with 5 random seeds to account for the randomness of the NMF algorithm. Across all 360 instances of the (producers, dimension, seed) configuration, the softmax content-serving rule with temperatures $\tau \in \{100, 10, 1, 0.1\}$ always converges to a unique Nash Equilibrium (NE) on the Movielens-100k dataset. Even with a low softmax temperature of $\tau = 0.01$, Algorithm 1 still converges to a unique NE in a large number of instances. Additionally, for the top-$k$ softmax serving rule reducing the top-$k$ value—making the serving rule "greedier"—results in fewer converged instances. The linear-proportional serving rule converges in many instances, and the round-robin serving rule as highlighted in Remark 2.2 always converges to the dimension with maximum weight.

| Serving Rule | $\tau = 100$ | $\tau = 10$ | $\tau = 1$ | $\tau = 0.1$ | $\tau = 0.01$ |
|---|---|---|---|---|---|
| (Full) Softmax | 360 | 360 | 360 | 360 | 342 |
| (Top-20) Softmax | 253 | 247 | 232 | 182 | 326 |
| (Top-10) Softmax | 170 | 171 | 165 | 147 | 299 |

(a) Converged instances for Softmax-based serving rules

| Serving Rule | Converged |
|---|---|
| Greedy | 269 |
| Linear | 357 |
| Round-Robin | 360 |

(b) Converged instances for temperature-independent rules.

Table 1: Convergence to Nash Equilibrium (NE) across serving rules on the Movielens-100k dataset. (a) Softmax serving rules with varying temperature $\tau$ and top $k$ values, (b) Temperature-independent rules.

Further, we examine the rate of convergence of the best response dynamics in Algorithm 1 to a Nash Equilibrium. In Figure 1 we plot the number of iterations (averaged over 40 runs) of Algorithm 1 with increasing number of producers and across varying embedding dimensions on the Movielens-100k dataset. Figures 1a and 1b plot the number of iterations to convergence with the linear and the softmax content-serving rule. In Appendix D.1, we provide figures for the synthetic datasets and observe insights similar to Figure 1.

Based on the observations above, we conclude that Algorithm 1 empirically seems to be a reliable and computationally efficient heuristic to find pure Nash equilibria for our engagement game, with performance scaling well with the number of producers. In contrast, a naive brute-force approach enumerating all best responses takes exponential time over the number of producers.

---

[7]We do not provide theoretical guarantees on the existence of pure NE or convergence of dynamics: in fact, Algorithm 1 does not converge in a few instances (Table 3), confirming that our game is not what is called a "potential" game. When a game is not potential, existence of pure NE and convergence of dynamics are not guaranteed, and equilibrium existence certification and computation is generally a computationally hard problem. Heuristics are often the best that we can hope for.

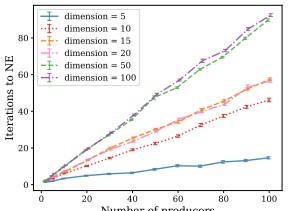 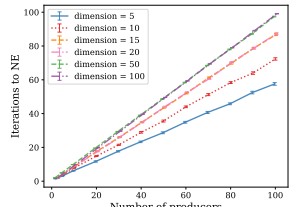

(a) the linear-proportional
serving rule Movielens-100k

(b) the softmax serving rule
Movielens-100k

Figure 1: Number of iterations of Algorithm 1 until convergence to a Nash Equilibrium on the Movielens-100k dataset. The different curves represent different embedding dimensions in the game $d \in \{5, 10, 15, 20, 50, 100\}$; the error bars represent standard error over 40 runs.

In the following, we study how changing the softmax temperature affects producer specialization and the producer utility. We consider 5 different values for the softmax temperature $\tau \in \{0.01, 0.1, 1, 10, 100\}$ and use the linear-proportional serving rule as a benchmark.

**Equilibrium results** In Figures 2 and 3, we highlight the impact of the content-serving rule and softmax temperature on the degree of specialization at the instance-level. We show a single (but representative) instance of the problem in each figure to provide a visual representation of producer specialization at equilibrium; our insights are consistent across our generated instances.

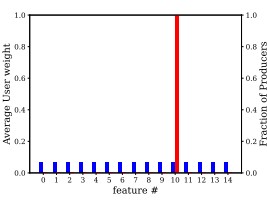 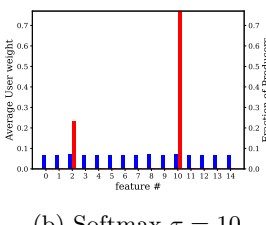 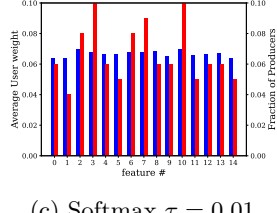 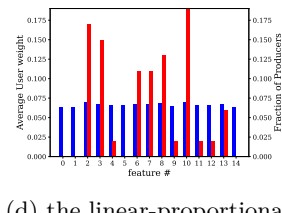

(a) Softmax $\tau = 100$     (b) Softmax $\tau = 10$     (c) Softmax $\tau = 0.01$     (d) the linear-proportional serving rule

Figure 2: Average user weight on each feature (blue, left bar) and fraction of producers going for each feature (red, right bar) $n = 100$ producers, embedding dimension $d = 15$. Lower softmax temperature leads to more producer specialization. User embeddings obtained from NMF on MovieLens-100k.

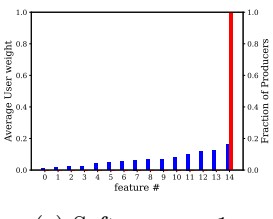 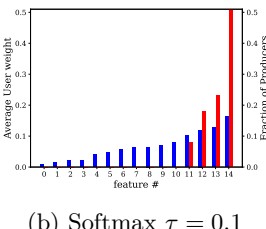 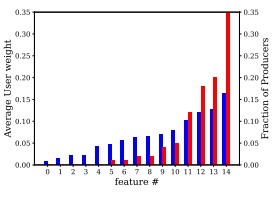 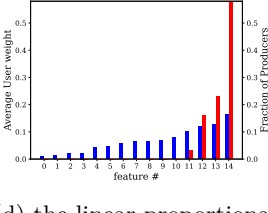

(a) Softmax $\tau = 1$     (b) Softmax $\tau = 0.1$     (c) Softmax $\tau = 0.01$     (d) the linear-proportional serving rule

Figure 3: Average user weight on each feature (blue, left bar) and fraction of producers going for each feature (red, right bar) $n = 100$ producers, embedding dimension $d = 15$. Lower softmax temperature leads to more producer specialization. Skewed-uniform distribution of users.

In Figure 2, we show the Nash equilibria for the linear-proportional serving rule and the softmax serving rule with varying temperatures on the MovieLens-100k dataset, with user embeddings obtained via NMF. For the softmax serving rule, we observe that the degree of specialization is decreasing in the temperature $\tau$. We see specialization occurring as the temperature drops from 100 to 0.01. A high temperature ($\tau = 100$) incentivizes

homogeneous content production: this is expected, as the content-serving rule becomes largely independent of producers' decisions (content is shown with probability converging to $1/n$), and producers maximize their utility by homogeneously targeting the highest-utility content ($\arg\max_{f\in d}\sum_{k\in K} c_k(f)$). When $\tau$ is of the order of 10 is when we first start seeing specialization; and the lowest temperature we experiment with ($\tau = 0.01$) leads to the most specialization.

In Figure 3, we replicate our experiment on the skewed-uniform dataset. Here too, the degree of producer specialization is decreasing in the softmax temperature. However, the levels of specialization seem to decrease for the skewed-uniform dataset and specialization seems to first occur at a lower temperature. This can be explained by the fact that low-weight features may not be worth targeting and ignored by the producers.

Further, we note that with the linear-proportional serving rule we observe a high level of specialization on the Movielens-100k and skewed-uniform datasets as seen in Figures 2d and 3d respectively.

In Appendix D.2, we present producer distribution figures for the uniform dataset, which exhibit insights similar to those in Figure 2. Additionally, we provide extended figures with a few additional softmax temperatures for the Movielens-100k and Skewed datasets (see Figures 5 and 10). Appendix E includes producer distribution figures for the sparse synthetic dataset.

**Producer utility at equilibrium** In Figure 4a, we plot the average producer utility with increasing softmax-serving temperature and with varying numbers of producers at a Nash Equilibrium. We observe that the producer utility is decreasing with temperature, and temperature $\tau = 0.01$ (near-hardmax) has the highest utility. We believe this provides an argument in favour of using low temperatures in the softmax content-serving rule. Recall that since the average user utility is identical to the average producer utility (up to a multiplicative factor), the benefit of a low temperature in the softmax-serving also extends to user utilities.

In Figure 4b, we compare the average producer utility across different serving rules while varying the softmax temperature. For softmax serving, a lower temperature leads to higher producer utility, similarly, in the top-$k$ softmax rules, reducing $k$ results in higher utility at the same temperature. This indicates that "greedier" serving strategies improve producer utility. However, there is a trade-off between increasing producer utility by making the serving rule greedier and the potential for decreased convergence to a Nash Equilibrium, as observed in the Nash convergence table 1. Note that the linear and round-robin serving rules are independent of temperature; we plot the same mean and standard error across all temperature values. Among these, round-robin serving yields the lowest producer utility, while linear serving remains competitive.

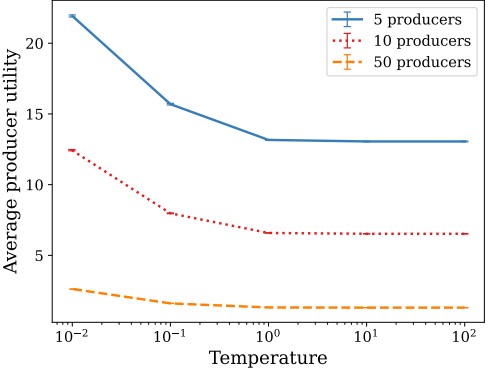

(a) Producer utility with varying producers, softmax serving

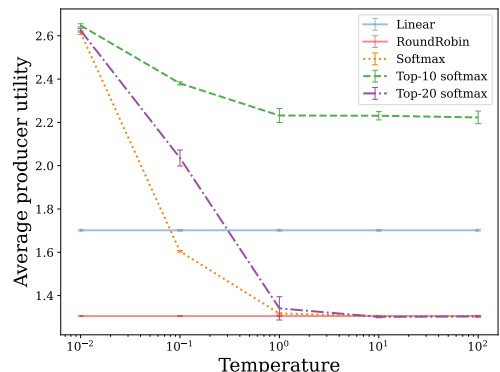

(b) Producer utility across serving rules, $n = 50$ producers

Figure 4: Average producer utility on the Movielens-100k dataset. (a) Varying the number of producers $n \in \{5, 10, 50\}$ with embedding dimension $d = 15$. (b) Comparing producer utility across serving rules: Linear (blue), RoundRobin (red), Softmax (orange), Top-10/20 Softmax (green/purple) with $n = 50$ producers and $d = 15$. Error bars represent standard error over 5 seeds.

# 6 Conclusion

In this paper, we studied engagement games, a game-theoretic model of producers competing for user engagement in a recommender system. Our main structural result showed that each producer targets a single feature in embedded space at equilibrium. We then leveraged this structural result to study content-specialization by producers and showed both theoretically and via extensive experiments that specialization arises at a pure Nash Equilibrium of our game, and as a result of natural game dynamics. We also observe that lower temperatures in the softmax content-serving rule incentivize specialization and improve producer utility. Our linear-proportional serving rule serves as competitive benchmark still demonstrating high levels of specialization and producer utility.

**Limitations**   As in previous work on producer competition in recommender systems, we assume that each producer is rational, and fully controls placement of their strategy $s_i$. Rationality is a common assumption in systems with strong profit motives for the producers. Full control may be less realistic, as producers can modify content features, but they do not know exactly how these changes affect the content embedding. However, this model provides a first-order approximation to real-life behavior, and the same assumptions are typical in related work (Hron et al., 2022; Jagadeesan et al., 2022; Yao et al., 2023a; Hu et al., 2023).

Our main structural result holds only for *non-negative embeddings*. While we make no assumption on how these non-negative embeddings are obtained, an interesting direction for future work is to study engagement games with potentially negative embeddings. There, it is not clear whether our best-response dynamics still converge, and propose to study no-regret dynamics as a direction for equilibrium computation.

Finally, we caution treating the equilibria of our engagement games as definitive; we rather present them as insights to competition in recommender systems, given the significant complexities of real-world recommender systems and environments in which they operate.

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

# A    Extended comparison with related work

| Paper | Producer Reward | Producer Dynamics? | User Dynamics? | Producer Equilibrium |
|---|---|---|---|---|
| Ours | Engagement | Yes | No | Pure NE |
| Ben-Porat et al. (2019b; 2020) | Exposure | Yes | No | Pure NE |
| Hron et al. (2022) | Exposure | Yes | No | Local NE |
| Jagadeesan et al. (2022) | Exposure | Yes | No | Mixed NE |
| Hu et al. (2023) | Mechanism design | Yes | No | Mixed NE |
| Yao et al. (2023a) | Engagement | Yes | No | CCE |
| Yao et al. (2023b; 2024) | Mechanism design | Yes | No | Local NE |
| Dean and Morgenstern (2022) | N/A | N/A | Yes | N/A |
| Lin et al. (2024) | Engagement | Yes | Yes | N/A |

Table 2: Comparing our paper to related work

Ben-Porat et al. (2019a; 2020) study exposure games with finite strategy spaces (producers selecting content from a finite catalog) and search for Pure NE using graph algorithms. Hron et al. (2022) focus on the concept of $\varepsilon$-local Nash Equilibria: i.e., given a joint strategy profile $(s_i, s_{-i})$, each producer's $i$ strategy $s_i$ is optimal in the open ball at $(s_i, s_{-i})$ with radius $\varepsilon$—this is a weaker notion of equilibrium than pure NE, which holds globally. The work of Jagadeesan et al. (2022) characterizes the support of the mixed Nash Equilibrium of their game, but do not provide a closed-form expression for said mixed NE—we instead provide a closed-form characterization in the restricted setting of Section 4. Hu et al. (2023) builds on the exposure model in Jagadeesan et al. (2022) but studies the mechanism design problem with the platform modeled by a linear contextual bandit. Yao et al. (2023a) focus on engagement games with the top-k serving rule, studying no-regret dynamics and the Coarse Correlated Equilibria (CCE) to which these dynamics converge. Yao et al. (2023b; 2024) address the mechanism design problem, aiming to incentivize welfare-maximizing local Nash equilibria. The works of Dean and Morgenstern (2022); Lin et al. (2024) model user dynamics, studying user polarization and do not model producer competition and equilibrium.

# B    Omitted Proofs

## B.1    Proof of Lemma 4.2

First, we note that producer $i$'s utility is convex in $s_i$ (following the same proof as in Appendix 3.1), however is generally not *strictly* convex. This implies (as for the proof in Appendix 3.1) that each producer $i$ has a best response in the standard basis $\mathcal{B}$. There may however be best responses supported outside of $\mathcal{B}$, and not exist an equilibrium supported on $\mathcal{B}$. Yet, note that if there exists a strategy profile in which each producer's strategy is supported on $\mathcal{B}$ and all producers best respond to each other, it is in fact an equilibrium supported on $\mathcal{B}$: no producer can improve their utility by deviating within $\mathcal{B}$, and no producer who cannot improve by deviating on $\mathcal{B}$ can improve by deviating outside of $\mathcal{B}$ since there is a best response on $\mathcal{B}$. In turn, there exists an equilibrium supported on $\mathcal{B}$ *if and only if* there exists an equilibrium when producers' strategies are restricted to $\mathcal{B}$.

Now, let us restrict ourselves to producers playing on $\mathcal{B}$. Pick any $f$ such that $n_f > 0$. We start by computing the utility of producer $i$ who plays $e_f$:

$$u_i(e_f, s_{-i}) = \sum_{k=1}^{K} p_i(c_k, e_f, s_{-i}) \cdot c_k^\top e_f = \sum_{k=1}^{K} \frac{\mathbb{1}[c_k = e_f]}{n_f}$$

where the second equality follows from the fact that $p_i(e_f, e_f, s_{-i}) \cdot e_f^\top e_f = \frac{1}{n_f}$ and $p_i(e_{f'}, e_f, s_{-i}) = 0$ for $f' \neq f$. Therefore, the engagement utility from user $c_k$ is 0 if $c_k \neq e_f$, and $1/n_f$ if $c_k = e_f$. Since there is a number $m_f$ of users with $c = e_f$, it follows that the total engagement utility for producer $i$, if it picks $s_i = e_f$, is given by $m_f/n_f$.

Now, let us study the deviations for any given producer $i$ with $s_i = e_f$. Suppose the producer deviates to $e_{f'}$ where $f' \neq f$. Then, the new number of producers picking $f'$ is $n_{f'} + 1$, and producer $i$ now obtains utility $m_{f'}/(n_{f'} + 1)$. Therefore, producer $i$ does not deviate if and only if for all $f'$, we have

$$\frac{m_f}{n_f} \geq \frac{m_{f'}}{n_{f'} + 1}.$$

Then $(n_1, \ldots, n_d)$ is an equilibrium iff no producer wants to deviate, i.e iff for all $f, f'$ such that $n_f > 0$,

$$\frac{n_f}{m_f} \leq \frac{n_{f'}}{m_{f'}} + \frac{1}{m_{f'}}.$$

If $n_f = 0$, there is no deviation for any producer that picks $e_f$ as such producers do not exist, and the inequality trivially holds. This concludes the proof.

## C Supplementary results on the Movielens-100k dataset

### C.1 Producer distribution

Figure 5 shows the producer distribution for the Movielens-100k dataset with softmax serving, illustrating increased specialization as the temperature decreases. Similarly, Figure 6 shows similar insights for top-20 softmax serving, however, Unlike full softmax, top-20 serving shows specialization even at high temperatures ($\tau = 100$), as it retains the top 20 producers and serves those producers almost randomly (high $\tau$). Figure 7 presents producer distributions for the three non-temperature-dependent rules: greedy, linear, and round-robin. Greedy and linear serving result in producer specialization, while round-robin, as highlighted in remark 2.2), leads all producers to target the highest-weight feature.

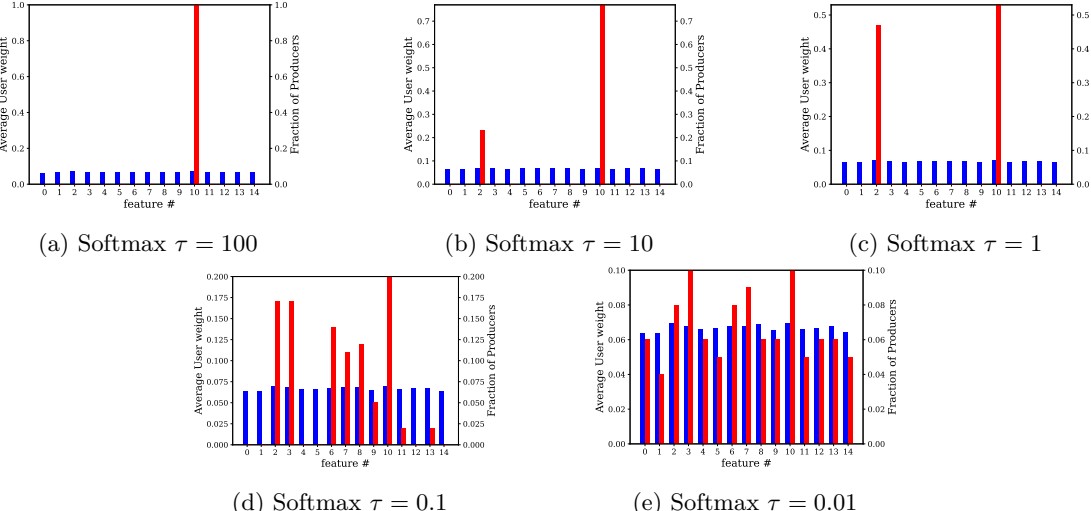

(a) Softmax $\tau = 100$      (b) Softmax $\tau = 10$      (c) Softmax $\tau = 1$

(d) Softmax $\tau = 0.1$      (e) Softmax $\tau = 0.01$

Figure 5: (Full) Softmax serving: Average user weight on each feature (blue, left bar) and fraction of producers going for each feature (red, right bar) $n = 100$ producers, embedding dimension $d = 15$. User embeddings obtained from NMF on MovieLens-100k.

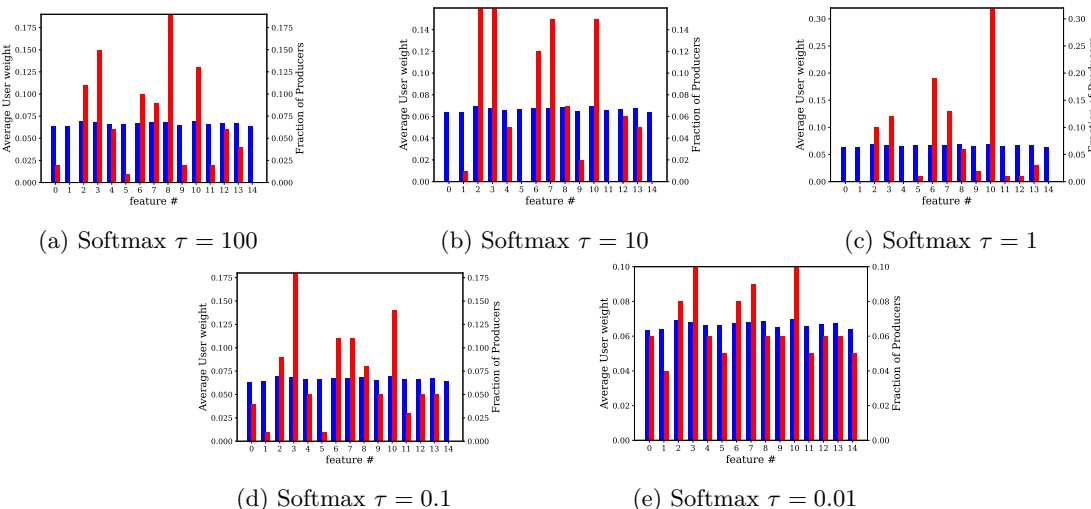

(a) Softmax $\tau = 100$      (b) Softmax $\tau = 10$      (c) Softmax $\tau = 1$

(d) Softmax $\tau = 0.1$      (e) Softmax $\tau = 0.01$

Figure 6: (Top-20) Softmax serving: Average user weight on each feature (blue, left bar) and fraction of producers going for each feature (red, right bar) $n = 100$ producers, embedding dimension $d = 15$. User embeddings obtained from NMF on MovieLens-100k, Note: unlike full softmax, top-20 serving shows specialization even at a high temperature.

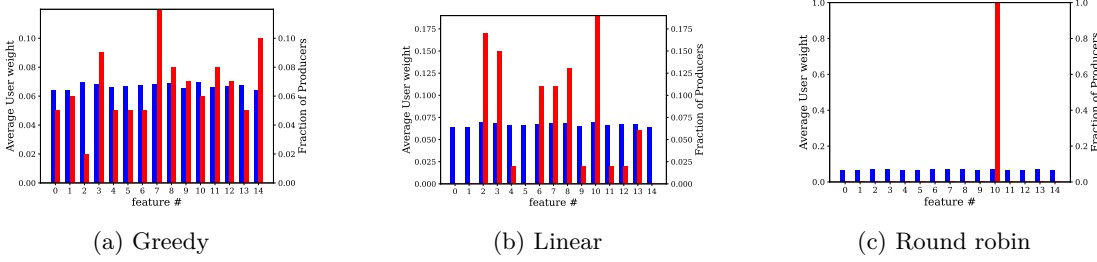

(a) Greedy      (b) Linear      (c) Round robin

Figure 7: Temperature independent serving rules: Greedy, Linear and Round-robin serving, Average user weight on each feature (blue, left bar) and fraction of producers going for each feature (red, right bar) $n = 100$ producers, embedding dimension $d = 15$. User embeddings obtained from NMF on MovieLens-100k.

# D Supplementary results on the uniform and skewed synthetic datasets

## D.1 Number of iterations until convergence

In Figure 8, we plot the number of iterations (averaged over 40 runs) of Algorithm 1 with increasing number of producers and across varying embedding dimensions. We do so on the Uniform and Skewed-uniform datasets. With both the linear-proportional and softmax content serving rules, we observe that Algorithm 1 seems to scale linearly in the number of producers, further highlighting the computational efficiency of our heuristic.

## D.2 Producer distribution

Figure 9 and Figures 10 provide producer distribution plots for softmax serving on the synthetic uniform and skewed datasets. These plots further highlight how the degree of specialization increases as the temperature decreases, over a few more softmax temperatures ($\tau = 10$ and $\tau = 0.1$).

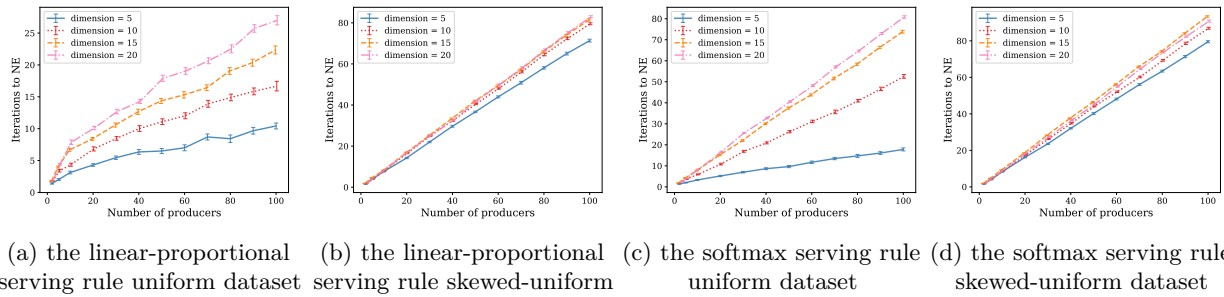

(a) the linear-proportional serving rule uniform dataset

(b) the linear-proportional serving rule skewed-uniform dataset

(c) the softmax serving rule uniform dataset

(d) the softmax serving rule skewed-uniform dataset

Figure 8: Number of iterations of Algorithm 1 until convergence to a Nash Equilibrium on the uniform, skewed-uniform datasets. The different curves represent different embedding dimensions in the game $d \in \{5, 10, 15, 20\}$; the error bars represent standard error over 40 runs.

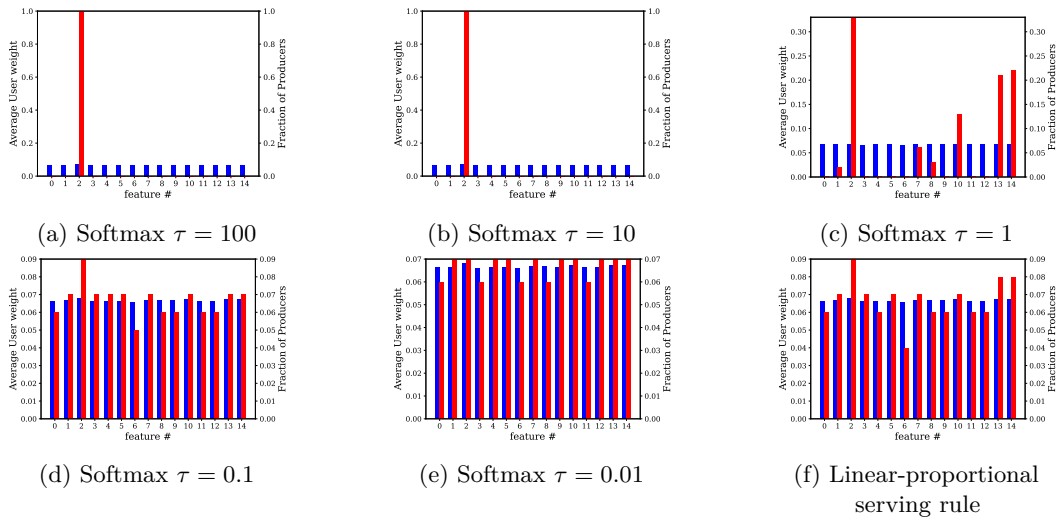

(a) Softmax $\tau = 100$

(b) Softmax $\tau = 10$

(c) Softmax $\tau = 1$

(d) Softmax $\tau = 0.1$

(e) Softmax $\tau = 0.01$

(f) Linear-proportional serving rule

Figure 9: Average user weight on each feature (blue, left bar) and fraction of producers going for each feature (red, right bar) $n = 100$ producers, embedding dimension $d = 15$. Lower softmax temperature leads to more producer specialization. Uniform distribution of users.

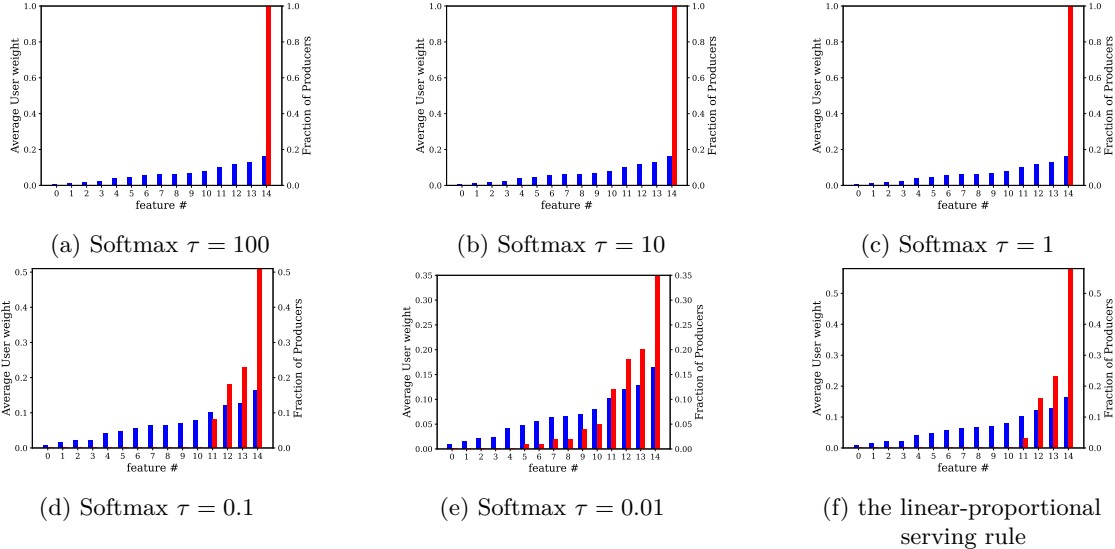

(a) Softmax $\tau = 100$

(b) Softmax $\tau = 10$

(c) Softmax $\tau = 1$

(d) Softmax $\tau = 0.1$

(e) Softmax $\tau = 0.01$

(f) the linear-proportional serving rule

Figure 10: Average user weight on each feature (blue, left bar) and fraction of producers going for each feature (red, right bar) $n = 100$ producers, embedding dimension $d = 15$. Lower softmax temperature leads to more producer specialization. Skewed-uniform distribution of users.

# E Experiments on the Sparse synthetic dataset

## E.1 Average producer utility

Figure 11 illustrates the average producer utility across serving rules for the sparse synthetic dataset, revealing trends similar to those observed in Figure 4b in the main text. Specifically, lower $k$ values and lower temperatures lead to higher producer utility; however, this comes with a decreased likelihood of convergence to a Nash equilibrium.

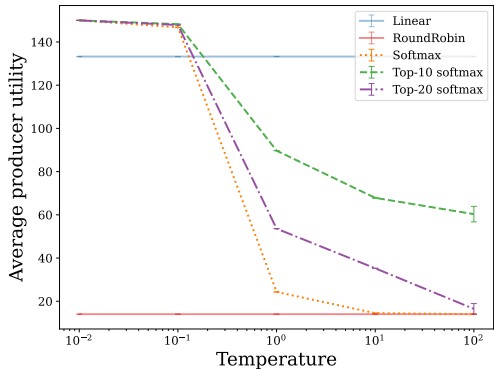

Figure 11: Average producer utility on the Sparse Synthetic dataset: comparing producer utility across serving rules: Linear (blue), RoundRobin (red), Softmax (orange), Top-10/20 Softmax (green/purple) with $n = 50$ producers and $d = 15$. Error bars represent standard error over 5 seeds.

## E.2 Producer distribution at Nash Equilibrium

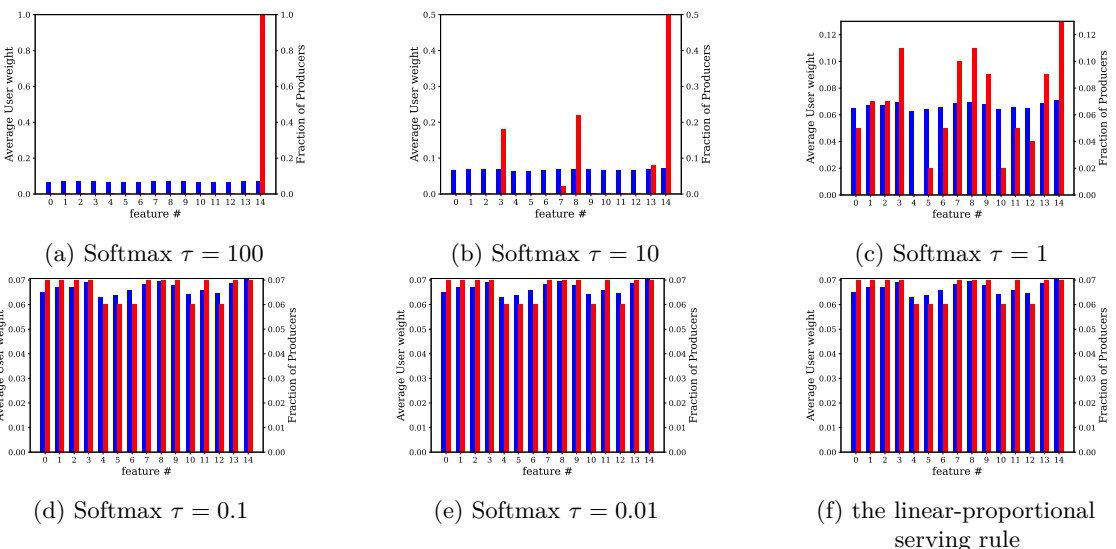

Figure 12: Average user weight on each feature (blue, left bar) and fraction of producers going for each feature (red, right bar) $n = 100$ producers, embedding dimension $d = 15$. Lower softmax temperature leads to more producer specialization. Sparse distribution of users.

# F Experiments on the AmazonMusic and RentTheRunway datasets

## F.1 Number of iterations till convergence

Figure 13 plots the number of iterations of Algorithm 1 until convergence to NE, averaged over 40 runs of best response dynamics, on the AmazonMusic and RentTheRunway datasets. Similar to Figure 1, we note a fast time to convergence that seems to scale linearly in the number of producers.

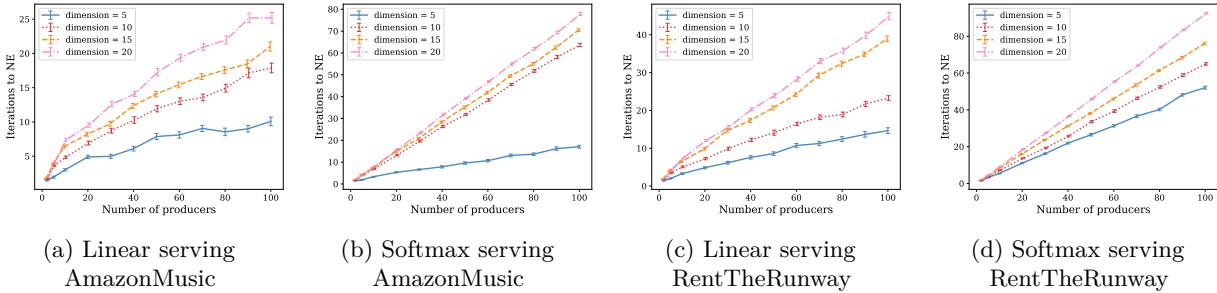

(a) Linear serving AmazonMusic

(b) Softmax serving AmazonMusic

(c) Linear serving RentTheRunway

(d) Softmax serving RentTheRunway

Figure 13: Number of iterations in Algorithm 1 until convergence to a NE on the AmazonMusic and RentTheRunway datasets. The different curves represent different embedding dimensions in the game $d \in \{5, 10, 15, 20\}$, error bars represent standard error over 40 runs.

## F.2 Producer distribution at Nash equilibrium

Figures 14 and 15 provide plots for the producer distribution at Nash equilibrium on the AmazonMusic and RentTheRunway dataset respectively. We note that our insights on how the softmax temperature affects specialization at equilibrium also arise in these two additional datasets: namely, lower temperatures lead to higher degrees of specialization.

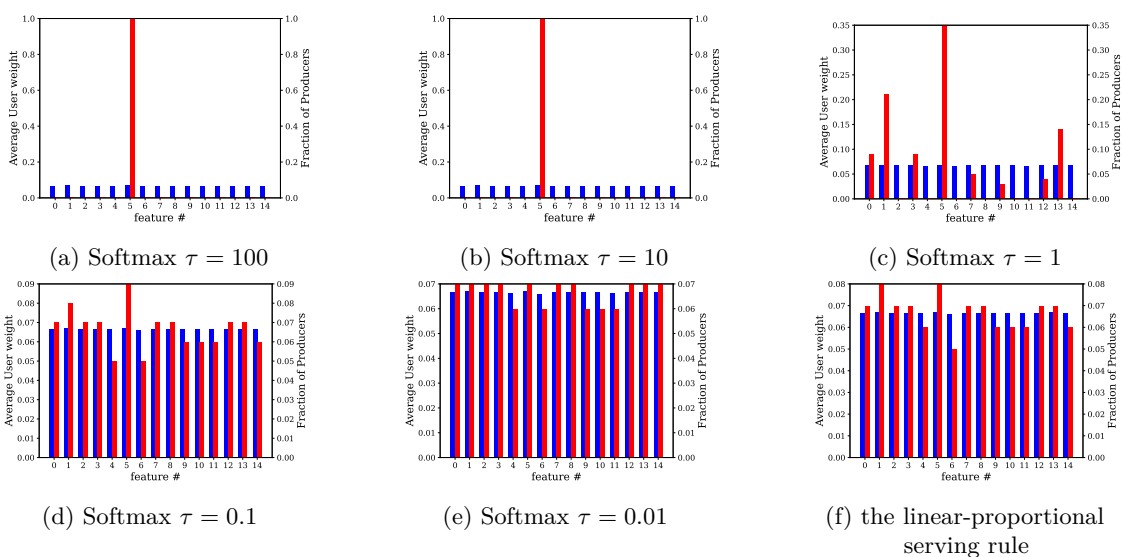

(a) Softmax $\tau = 100$

(b) Softmax $\tau = 10$

(c) Softmax $\tau = 1$

(d) Softmax $\tau = 0.1$

(e) Softmax $\tau = 0.01$

(f) the linear-proportional serving rule

Figure 14: Average user weight on each feature (blue, left bar) and fraction of producers going for each feature (red, right bar) $n = 100$ producers, embedding dimension $d = 15$. Lower softmax temperature leads to more producer specialization. User embeddings obtained from NMF on the AmazonMusic dataset.

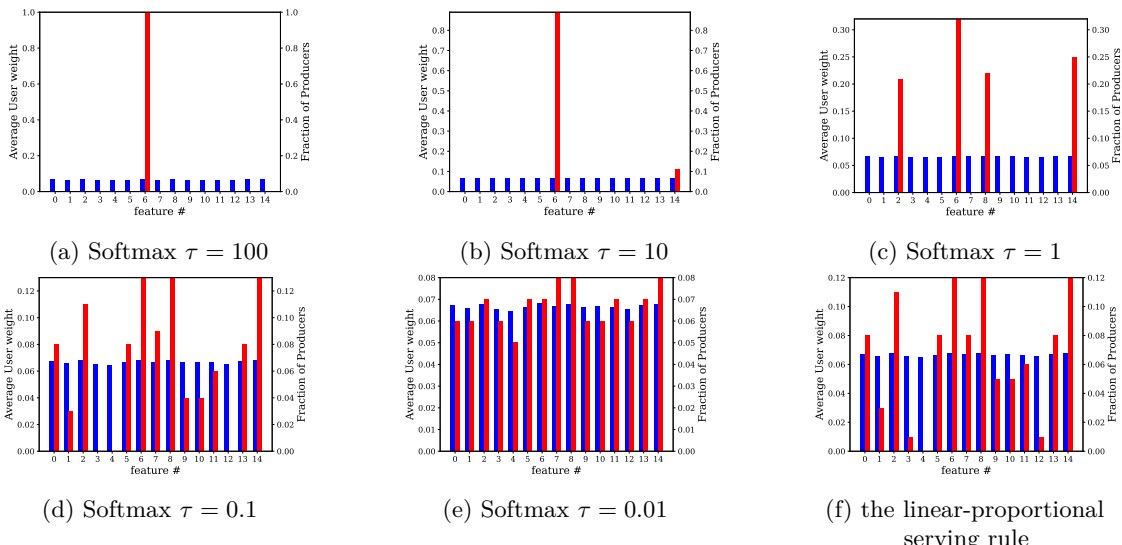

(a) Softmax $\tau = 100$     (b) Softmax $\tau = 10$     (c) Softmax $\tau = 1$

(d) Softmax $\tau = 0.1$     (e) Softmax $\tau = 0.01$     (f) the linear-proportional serving rule

Figure 15: Average user weight on each feature (blue, left bar) and fraction of producers going for each feature (red, right bar) $n = 100$ producers, embedding dimension $d = 15$. Lower softmax temperature leads to more producer specialization. User embeddings obtained from NMF on the RentTheRunway dataset.

### F.3 Average producer utility

In Figure 16, we plot the average producer utility with increasing softmax-serving temperature and with varying numbers of producers on the AmazonMusic and RentTheRunway datasets. As in Figure 4a, we observe that the producer utility is decreasing with temperature, and temperature $\tau = 0.01$ (near-hardmax) has the highest utility. This further supports using low temperatures in the softmax content-serving rule.

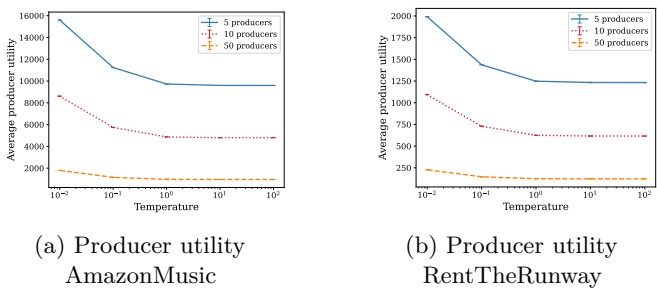

(a) Producer utility AmazonMusic     (b) Producer utility RentTheRunway

Figure 16: Average producer utility is decreasing in the softmax temperature, results on the AmazonMusic and RentTheRunway datasets. The different curves represent different number of producers in the game $n \in \{5, 10, 50\}$, embedding dimension $d = 15$, error bars represent standard error over 5 seeds

In Figure 17 we compare the average producer utility with the softmax serving rule (across increasing temperatures) v.s with the linear-proportional serving rule. We observe that across both datasets, the lowest softmax temperature we experiment with ($\tau = 0.01$) leads to a greater utility when compared to linear serving. However, linear serving still obtains a competitive utility, greater than that with softmax temperatures $\tau \in \{0.1, 1, 10, 100\}$.

## G Tables for Nash Equilibrium convergence

For our engagement game, we consider 12 different number of producers $(2, 5, 10, 20, \ldots, 100)$ and 4 different embedding dimensions $(5, 10, 15, 20)$. Each of these $12 \times 4$ games are instantiated with 5 random seeds for the

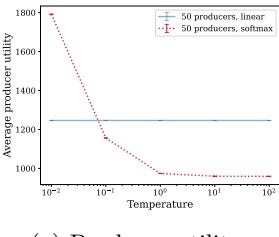
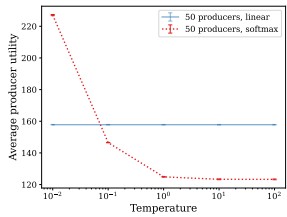

(a) Producer utility
AmazonMusic

(b) Producer utility
RentTheRunway

Figure 17: Average producer utility with linear serving (in blue) v.s with softmax-serving (in red) across increasing temperatures on the AmazonMusic and RentTheRunway datasets. $n = 50$ producers, embedding dimension $d = 15$, error bars represent standard error over 5 seeds.

randomness in the draws of the synthetic data, and the randomness of NMF algorithm on the real datasets. We observe that on each of these 240 instances of (#producer, dimension, seed), the softmax content-serving rule with temperatures $\tau \in \{100, 10, 1, 0.1\}$ *always* converges to a unique NE on all the datasets. With softmax temperature $\tau = 0.01$, Algorithm 1 still converges to a unique NE in a large number of instances. The linear-proportional serving also converges to a unique NE almost always.

| Dataset \ Serving Rule | Linear | Softmax | | | | |
|---|---|---|---|---|---|---|
| | | $\tau = 100$ | $\tau = 10$ | $\tau = 1$ | $\tau = 0.1$ | $\tau = 0.01$ |
| Uniform | 239 | 240 | 240 | 240 | 240 | 232 |
| Skewed-uniform | 240 | 240 | 240 | 240 | 240 | 240 |
| Movielens-100k | 240 | 240 | 240 | 240 | 240 | 233 |
| AmazonMusic | 240 | 240 | 240 | 240 | 240 | 236 |
| RentTheRunway | 240 | 240 | 240 | 240 | 240 | 237 |

Table 3: Number of instances in which Algorithm 1 converges to a Nash equilibrium

## H   Reproducibility

Here we briefly describe the datasets, embedding generation, and experiments. The code and further details are available in the supplementary.

**Datasets:**   We import the Movielens-100k dataset from the `scikit-surprise` package. For the Amazon-Music ratings we use the "ratings only" `Digital_Music.csv` from `https://nijianmo.github.io/amazon/index.html`, this dataset has approximately 1.5 million ratings, $840k$ unique users and $450k$ unique items. For RentTheRunway we use `https://cseweb.ucsd.edu/~jmcauley/datasets.html#clothing_fit`, this dataset has around $190k$ ratings, $100k$ unique users and $5.8k$ unique items.

**Embedding generation:**   We consider the following embedding seeds $\{13, 17, 19, 23, 29\}$ for randomness in the embedding generation. These 5 embeddings seeds are used in the random draws for the synthetic uniform and skewed embeddings, and for the randomness in the Non-negative matrix factorization embeddings for the Movielens-100k, AmazonMusic and RentTheRunway datasets. Note that the real data embeddings are generated using the NMF implementation in `scikit-surprise` where we pick 4 different factors $d \in \{5, 10, 15, 20\}$. The synthetic and real data embeddings take 2.6 CPU hours in total to generate and are saved for further use in the experiments.

Note that to parallelize workloads for embedding generation and for the following experiments we use Slurm job arrays.

**Experiment 1 : Convergence of Algorithm 1**   For our engagement game, we consider 12 different number of producers $n \in \{2, 5, 10, 20, 30, 40, 50, 60, 70, 80, 90, 100\}$ and 4 different embedding dimensions $d \in \{5, 10, 15, 20\}$. Each of these $12 \times 4$ engagement games are instantiated with the 5 embedding seeds described above which determine the user embeddings for a dataset. We then run Algorithm 1 40 times with the maximum number of iterations $N_{max}$ set to 500. In each of the 40 trials of Algorithm 1 the producer strategies are randomly initialized using the sequential seeds $\{1, 2, \ldots 40\}$. We then plot the mean and standard error across these in Figure 1,8 and 13. All plots use a softmax temperature of $\tau = 1$ for simplicity of exposition.

This experiment on the Uniform, skewed and Movielens-100k datasets take approximately $50, 74$ and 3 total CPU hours respectively. With the larger scale AmazonMusic and RentTheRunway it takes $\sim 4.5$k and 1k total CPU hours.

**Experiment 2 : Producer distribution and utility**

**Producer distribution:**   Here we fix the embedding seed to 17, and plot the producer distribution for the linear serving rule and for the softmax-serving rule after Algorithm 1 terminates. Note that here we don't report mean producer distribution across embedding seeds as this can hide lack of specialization, in the following manner: for a given embedding, only a few features are targeted by the producers, like in Figure 9b all producers go for index #2. However, the specific index of the feature could change as the embedding seed changes. In that case, when averaging over the seeds, we may be under the impression that specialization occurs, when it does not.

**Producer utility:** Here for each of the $12 \times 4$ instances of #producers $n$ and dimension $d$, we instantiate the engagement game with linear-serving and 5 different softmax-serving temperatures $\tau \in \{0.01, 0.1, 1, 10, 100\}$. We measure the average producer and user utility at the unique Nash equilibrium and report its mean across the 5 embedding seeds. Average utilities are measured by taking the average across converging runs, and non-converging seeds are dropped.

This experiment on the Uniform, skewed and Movielens-100k datasets take approximately $4.4, 7$ and 0.5 total CPU hours respectively. With the larger scale datasets, AmazonMusic and RentTheRunway, it takes $\sim 530$ and 105 total CPU hours.

