# OpenReview forum: "Producers Equilibria and Dynamics in Engagement-Driven Recommender Systems"
_TMLR — Accepted by TMLR_

### Review · Reviewer_K3oA · 2024-12-03

**Summary Of Contributions:**

This paper investigates the equilibrium behavior of content producers in engagement-driven recommender systems. The authors propose a game-theoretic model where producers aim to maximize user engagement (rather than exposure, as many previous works have instead considered), considering two content-serving rules: the softmax and linear-proportional rules. They derive theoretical results (under key simplifying assumptions) showing that at equilibrium, producers specialize by focusing on a single feature in the embedded content space, driven by competition for engagement.

Using synthetic and real-world datasets, the authors examine these results experimentally.  Their findings demonstrate the emergence of producer specialization, and examine its dependence on the particular content-serving rule and softmax temperature. The paper also introduces a heuristic algorithm to compute pure Nash equilibria, which is shown to converge efficiently in most cases. The experimental results reveal that lower softmax temperatures promote specialization and maximize both producer and user utilities. These findings provide insights into how platform design choices influence content diversity and engagement outcomes.

**Audience:**

Yes

**Broader Impact Concerns:**

No concerns.

**Claims And Evidence:**

Yes

**Requested Changes:**

- While it may not be feasible to explore theoretically, it would be very interesting to investigate the robustness of the equilibrium results when Assumptions 3.1 (strictly positive user preferences) and 3.2 (diverse user preferences) are relaxed. For instance, simulate scenarios with sparse or noisy user embeddings to assess whether specialization and equilibrium still hold.
- It would be useful to provide some analysis of the failure cases where Algorithm 1 does not converge to a Nash equilibrium.
- Though more of a stretch from the current version of the paper, it could be valuable to empirically investigate alternative content-serving rules beyond softmax and linear-proportional, to better understand their impact on producer specialization and engagement outcomes.

**Strengths And Weaknesses:**

**Strengths**

- The focus of the paper on engagement (as opposed to exposure) is more aligned with real-world platforms like YouTube and Instagram.
- The authors provide a clear structural characterization of producer equilibrium under stated assumptions, proving that producers specialize in a single feature in the embedded space at equilibrium.
- The experiments utilize both synthetic and real-world datasets (e.g., MovieLens-100k, Amazon Music, Rent the Runway), providing diverse evidence to support the theoretical claims.
- Overall, the characterization of how specialization naturally emerges in engagement-driven games is a valuable contribution, and once again aligns on a high level with my perceptions of the real-world dynamics of these platforms, making the results intuitive.

**Weaknesses**

- Key assumptions (e.g., strictly positive user preferences, fully diverse user embeddings, single-minded users) limit the practical takeaways of the results.
- The heuristic algorithm does not guarantee convergence to a Nash equilibrium in all cases, particularly under extreme configurations such as very low softmax temperatures.
- The experiments do not model noisy or adversarial user behaviors, which are present and critical to address in real-world recommender systems. Similarly, dynamic user and producer behaviors are not considered.
- The model does not account for heterogeneity in producers' abilities to create different types of content, which is a significant factor in real-world platforms and could substantially impact equilibrium outcomes.
- While the paper focuses on softmax and linear-proportional rules, it does not explore other plausible serving rules that might yield different equilibria or engagement patterns.
- While the paper offers some discussion on practical implications, it does not offer much actionable advice on how platforms could use these findings to mitigate undesirable outcomes like homogenization or clickbait prevalence.

---

> ### Author Response · Authors · 2025-01-16
>
> We thank the reviewer for the detailed comments and constructive feedback, and address the requested changes below.
>
> **Relaxing the Assumptions**
>
> We appreciate the reviewer’s suggestion to consider sparse features.We have incorporated this and constructed a sparse synthetic dataset as follows: user embeddings $c\in \mathbb{R}^d$ are generated uniformly at random over the probability simplex and then subjected to an element-wise masking operation. This operation applies random Boolean vectors with 90% of the values set to zero. The experimental results for this dataset are included in Appendix E, where we observe a similar trend: producer specialization increases as the softmax temperature decreases (Figure 12). The average producer utility also follows a consistent trend, increasing with lower softmax temperatures.
>
> **Additional Serving Rules**
>
> We have incorporated the Top-$k$ softmax and round-robin serving rules as additional baselines:
> - Top-$k$ softmax: This rule selects the top-$k$ producers with the highest alignment for a user and serves them using a softmax distribution with temperature $\tau$.
> - Round-robin: This rule serves producers cyclically, where all users are first shown content from producer 1, then producer 2, and so on.
> Formal descriptions of these serving rules have been added to Section 2.
>
> We also highlight a new Figure 4b in Section 5, which plots the average producer utility across all serving rules for the Movielens-100k dataset. The key observation is that lower temperatures and smaller top-k values result in higher producer utility. However, this improvement comes at the cost of a reduced likelihood of convergence to Nash equilibrium, as shown in the updated Table 1. A similar plot is provided for the sparse dataset in Appendix E (Figure 11).
>
> **Algorithm 1 Non-Convergence Investigation**
>
> Thank you for suggesting this analysis. Upon investigation, we observed cycling behavior when Algorithm 1 fails to converge. Specifically, while the total producer utility remains approximately constant (even after reaching the maximum iteration count of 500 steps in our experiments), the producer with the highest utility alternates between steps. We are happy to include a plot illustrating the total producer utility over the steps of Algorithm 1 to further clarify this phenomenon.

---

> > ### Comment · Reviewer_K3oA · 2025-01-17
> > **reviewer reply**
> >
> > Thank you to the authors for incorporating my feedback and updating the draft.  I think the analysis on non-convergence would be interesting to include.  I am satisfied with the discussion and revision, and will suggest acceptance for this paper.

---

> > > ### Author Response · Authors · 2025-01-17
> > >
> > > Thank you for your support! We appreciate your feedback and will include this plot in the next revision.

---

### Review · Reviewer_DPx2 · 2024-12-17

**Summary Of Contributions:**

The paper investigates content producer incentives in recommender systems, building on a recent line of work from the machine learning community.

The paper analyzes a game-theoretic model for content producer competition facilitated by the recommender system. Consumer preferences and content are D-dimensional vectors. The recommender system uses a content-serving rule (i.e., linear proportional rule, or a softmax rule) that is a continuous version of maximizing engagement. Each producer (strategically) chooses a D-dimensional content vector, and producer utility is based on engagement from the recommendations that the producer wins.

The paper analyzes the equilibria in the game between producers, providing conceptual insights about the equilibrium structure. The paper proves one general property of the equilibria --- that the equilibrium support is at the standard basis vectors, even when users are not necessarily located at the standard basis vectors --- and also characterizes the equilibria when the users are located at the standard basis vectors. The paper empirically analyzes equilibria on synthetic data and on the MovieLens-100K dataset, analyzing the role of the temperature parameter in the content-serving rule.

**Audience:**

Yes

**Claims And Evidence:**

Yes

**Requested Changes:**

Addressing the following questions/requests is critical to securing my recommendation for acceptance.
- Does the proof of Theorem 3.4 implicitly assume that the temperature is sufficiently high for the softmax rule? Please clarify this.
- The paper does not prove equilibrium existence for general sets of users, which makes Theorem 3.4 a bit hard to interpret. Does a mixed Nash equilibrium always exist by Glicksberg's theorem, because the utility functions are continuous? It would be helpful to clarify this.
- In Section 3, it seems like the assumption is that all of the features are strictly positive (Assumption 3.1), but the standard basis vectors in Section 4 do not seem to satisfy this assumption. This made it hard to compare the results in Sections 3 and 4.

Addressing the following questions is not critical to securing my recommendation for acceptance, but could strength the work in my view.
- The paper interchangeably uses "consumers" and "users". I think that it would be clear to stick to one of these.
- When does a pure strategy equilibrium exist for general sets of users? I might expect that pure strategy equilibria would stop existing when $\tau$ is sufficiently small (and this also appears to be the case in the experiments).
- What would happen if producers had to pay a cost for creating content that depends on the content? Would this change the insight that producers always target one dimension?

**Strengths And Weaknesses:**

Strengths:
- The paper makes a nice contribution to the content producer incentives literature, incorporating that the producer reward is based on engagement and that the content-serving rule is continuous. (I found the comparison in Appendix A very helpful.) This clean model gives rise to interesting economic behaviors.
- The paper clearly explains the conceptual insights emerging from the equilibrium structure analysis. The first conceptual insight (from Theorem 3.4 in Section 3) is that each producer targets a single type of content, even if consumers care about multiple types of content at once. The second conceptual insight (Lemma 4.2 in Section 3) is that the supply of content (up to rounding) matches consumer demand when consumers are single-minded (Section 4). The third conceptual insight is that low temperatures incentivize specialization and lead to higher producer utility (Section 5).
- The paper has a nice mixture of empirical and theoretical results.

Weaknesses:
- The proof of the main result in Section 3 (i.e., Theorem 3.4) seems to make an implicit assumption for the softmax rule: that the temperature is sufficiently large. Otherwise, the function f might not be strictly convex, and strict convexity is used in the proof. Is this true? (See my question in the requested changes section.)
- The main result in Section 3 (i.e., Theorem 3.4) analyzes the equilibrium structure assuming that equilibria exist, but does not establish equilibrium existence. This makes hard to interpret if there is a regime where equilibria do not exist.
- The theoretical equilibrium characterization results in Section 4 (i.e., Lemma 4.2) are restricted to the case where the users are located at the standard basis vectors. As the paper notes, this case corresponds to "single-minded" consumers who are only interested in one dimension of content. This theoretical analysis does not capture the more realistic case where consumers care about multiple dimensions of content.
- The model assumes that content production is free for producers, and that engagement and consumer utility are aligned. However, I think that these assumptions are fairly mild and also reasonable to make in a stylized model.
- The mathematical analysis in the paper is fairly straightforward.

---

> ### Author Response · Authors · 2025-01-16
>
> Thank you for your detailed review and thoughtful questions.
> Quick fix: we have updated the text to consistently use "users" throughout the manuscript.
>
> Below, we address the points you raised and provide clarifications where needed:
>
> **Temperature must be sufficiently high:**
>
> You are correct that the temperature must be sufficiently high, and while we state this condition in claim 3.6, we will explicitly add it to the theorem. The condition is that for a producer $i$ and a user $c$,
> $c^\top s_i \geq \tau \log(\sum_{j \neq i} c_k^\top s_j/\tau)$. Restricting ourselves to $\Vert s \Vert = 1$, as the argument remains unchanged, and letting $c_{min}$ denote the minimum possible value of a coordinate in $c$, we note that $c^\top s_i \leq 1$, This implies the following *sufficient* condition:
> $$
> \tau (1 - \log \tau) \geq 1/\log(c_{min}(n-1)).
> $$
>
> This condition may not be overly restrictive for larger games, but you are correct that it is an important requirement and will be added to the theorem statement. In particular, this provides an intuition of why the dynamics start breaking down at very low softmax temperatures—absent convexity, even computing the best response for a single producer (let alone a Nash equilibrium) becomes a difficult optimization problem; and, intuition of why temperatures that are too low may be undesirable, highlighting a trade-off between improved producers utilities and convergence/complexity of the problem.
>
> **Nash existence**:
>
> A first quick remark here is that our game is *not* a zero or constant-sum game. Although the probabilities sum to 1, the producer utilities do not sum to a constant amount due to the multiplication by the $c^\top s$ term for engagement.
>
> Regarding pure Nash equilibria:
> We note that a pure Nash equilibrium might not exist. Existence of pure Nash is often hard to characterize in non-zero-sum games. The typical argument is that the game admits a potential function, in which case a Nash equilibrium must exist. However, if a game has a potential function, Algorithm 1 (best-response dynamics) *must* converge to a pure Nash. Our experiments show that the best-response dynamics may not always converge, so we know our game is not a potential game. This does not rule out the existence of pure Nash, but it makes it hard to theoretically characterize existence.
>
> Regarding mixed Nash equilibria: note that while our game is not zero-sum, sufficient conditions for the existence of a mixed Nash equilibrium are continuity of the utilities and compactness of the strategy set S. In particular:
> 1) For the softmax serving rule, note that the induced utility is continuous. Further, the set S of vectors with $s \geq 0$ and $\Vert s \Vert \leq 1$ is compact. This is enough to show the existence of a mixed Nash.
> 2) For the linear-proportional serving rule, the argument is a tiny bit more involved, and is the main reason behind Assumption 3.1: if c’s are allowed to have 0 entries, then it can be the case that there exists (non-zero) strategies $(s_1,...,s_n)$ such that for all $i$, $c^\top s_i = 0$. At such a point, the linear-proportional serving rule is discontinuous (it is easy that it behaves inconsistently and converges to different limits from different directions). Basically, having 0 coordinates induces weird corner cases that technically break the worst-case theory.
>
> Now, under Assumption 3.1., the existence of a mixed Nash still holds. The first thing to note is that the linear serving rule utility function is increasing in $c^\top s_i$, with $c > 0$. This means that in particular, we can always strictly increase our utility by increasing $\Vert s_i \Vert$. Therefore, any $\Vert s_i \Vert < 1$ is dominated. Hence, we can restrict our search to $S’ = \{s \geq 0, \Vert s_i  \Vert =  1 \}$.
>
> Now, this space $S’$ is compact. Further, on $S’$, the linear-proportional utility is continuous (no non-zero strategy can lead to $c^\top s = 0$ when $c > 0$). This concludes the proof.
>
> We are happy to add a new lemma showing the above, and that a mixed Nash equilibrium always exists, in the next version of the paper!

---

> > ### Author Response · Authors · 2025-01-16
> >
> > **Strict positivity**:
> >
> > We acknowledge the mismatch you point out. Assumptions 3.1 and 3.2 are sufficient but not necessary for our purposes. These assumptions are here to guarantee a more general condition: in the linear serving case, at an equilibrium ($s_1^*,\ldots,s_n^*$), it is never the case that a user is not served (i.e., for a user c, we have $c^\top s_i^*$ are 0 for all producers i).
> >
> > We believe that this assumption is reasonable: it basically states that almost no user on the platform has preferences so niche that all producers ignore them at equilibrium. Further, it holds under Assumptions 3.1 + 3.2. Section 4 proves that it also holds under single-minded users (at equilibrium, the producers divide themselves proportionally, and so at equilibrium there must be a non-zero producer in that user's direction).
> >
> > The reason we did not state this as our main assumption is because it is a condition on the equilibrium directly, which is an endogenous quantity of the problem, not an assumption of the parameters of the problem. However, we would be happy to revise the paper to explicitly state this assumption, and highlight how both Assumptions 3.1 + 3.2, and Assumption 4.1 both satisfy this more general assumption.
> >
> > We also note that Section 4 is aimed at providing a simple setting where a theoretical characterization is possible, and to use this theoretical characterization to build some intuition on how much effort producers spend in each direction, and then see experimentally in Section 5 whether this simple intuition holds and when it breaks. We note that this is not central to the paper; the results of Section 4 are standalone and Assumption 4.1. Is *only* made in Section 4, not in 3 nor in the experiments.
> >
> > In comparison, Section 3 and the observation that it is sufficient to optimize on the unit basis is *central* to our algorithm and to our experiments. If the reviewer believes this would make the paper more coherent, we are happy to move Section 4 and its insights in the Appendix and present it as high-level intuition of the dynamics of our problem (features with more users see more weight from the producers).
> >
> > **Cost for content creation**
> >
> > We can incorporate a cost function $c(s): \mathbb{R}^d \to \mathbb{R}^+$ for content creation, where the producer’s utility is defined as the engagement utility minus the cost of production. Our theoretical results remain valid as long as $c(s)$ is concave. However, we acknowledge that addressing general cost functions remains an open problem.

---

> > > ### Comment · Reviewer_DPx2 · 2025-01-20
> > >
> > > Thanks to the authors for their detailed response! These clarifications addressed my concerns, and I'll recommend acceptance for this paper.

---

> > > > ### Author Response · Authors · 2025-01-20
> > > >
> > > > We deeply thank the reviewer for carefully reading our response and for their support. Thanks again!

---

### Review · Reviewer_w2au · 2024-12-18

**Summary Of Contributions:**

This paper investigates producer Nash Equilibria (NE) in recommendation systems. The proposed approach presents a theoretical case for producer specialization to maximize user engagement and presents an algorithm to obtain the NE. The algorithm is shown to converge on a synthetic dataset as well standard recommendation system datasets.

**Audience:**

Yes

**Claims And Evidence:**

No

**Requested Changes:**

In general, my outlook of the paper will be improved in terms of validating the contribution and ensuring the claims are verified if the authors address the weaknesses of the paper to strengthen the paper.

- Include reasonable baseline methods, e.g., some of the methods from Table 2 in the experiments and show how the proposed methods improves any reasonable metric such as total utility.

- Investigate scaling of the approach in asymptotic case as embedding dimension grows.

- Move Algorithm 1 to main text to improve readability. It is a core contribution of the paper and referenced in the main text, it cannot be left to appendix.

- Fix typos, e.g., Section 5.2 "softmax content-serving rule rule"

- Minor Question: The single mindedness of the users is an important assumption. How to evaluate this key assumption? Is there a procedure to test how well this works in specific dataset that the authors recommend?

**Strengths And Weaknesses:**

## Strengths
- Theoretical results consist of proofs with clear assumptions, technically sound derivations and compelling results. The proofs in the main paper are correct (checked in reasonable detail) whereas appendix proofs also seem correct but were checked less carefully.
- The application of game theoretic analysis for producer strategy in recommendation systems is a very interesting topic that is bound to resonate with at least part of the TMLR research community
- The paper is well motivated and reasonably easy to follow in the technical parts

## Weaknesses
The main problems with the paper are in the experimental evaluation.
- There seems to be a lack of comparison to baselines. Why not compare total utility obtained from the proposed technique for the various datasets with other methods or even a naive baseline? For example produce a figure similar to Figure 4 with the method as well some a selection of reasonable baselines.
- The scaling of the proposed algorithm in Figure 1 requires some further analysis. I understand that theoretical complexity calculation for the Nash Equilibrium can be challenging and the bounds quite loose in the worst case. However, it would be nice to see how a very large embedding dimension affects number of iteration needed. What if the dimension is 100? Hopefully the pattern of diminishing increases in the iteration count to convergence continues which would indicate a low asymptotic complexity, a good result for this approach. On the other hand, if the empirical complexity is poor perhaps an argument as to the inherently low dimensional embedding space covering user interests could be made.

---

> ### Author Response · Authors · 2025-01-16
>
> We thank the reviewer for their thoughtful review and address the requested changes below.
>
> Quick fixes: We agree with the suggestions and have moved Algorithm 1 to Section 3. We appreciate spotting the typo, which has been corrected.
>
> **Scaling of Algorithm 1 and Higher Dimensions**
>
> Figure 1 for the number of iterations to Nash equilibrium now includes larger embedding dimensions, specifically $d=50$ and $d=100$ , the scaling remains linear, indicating a low asymptotic complexity for Algorithm 1.
>
> **Comparison to baselines**
>
> We have incorporated the Top-k softmax and round-robin serving rules as additional baselines.
>
> - Top-$k$ softmax: This rule selects the top-$k$ producers with the highest alignment for a user and serves them using a softmax distribution with temperature $\tau$.
> - Round-robin: This rule serves producers cyclically, where all users are first shown content from producer 1, then producer 2, and so on.
> Formal descriptions of these serving rules have been added in Section 2.
>
> We also highlight a new Figure 4b, which plots the average producer utility across all serving rules. The key observation is that lower temperatures and smaller top-k values result in higher producer utility. However, this improvement comes at the cost of a reduced likelihood of convergence to Nash equilibrium, as seen in the updated Table 1.
>
> To provide further insights, we include the producer distributions for the Top-20 softmax, greedy, and round-robin rules in Appendix C (Figures 6 and 7). Notably, the Top-$k$ softmax rule shows a similar trend of increasing producer specialization with decreasing temperature. Unlike the full softmax rule, Top-$k$ exhibits specialization even at high temperatures (Figure 6a), as it retains the top-20 producers and serves them randomly (for high $\tau$), thereby maintaining an element of greediness in the serving rule.
>
> About comparison with Table 2, we clarify that the major difference is that producers are playing a different game. In much of the related work, producers are modeled as aiming to maximize exposure, i.e. the probability of their content being seen. In our work, however, we extend this model to note that producers may want to maximize engagement instead of exposure, directly taking into account how good of a fit their content is to users, noting that the better the fit, the more users engage with it. Effectively, the main difference is that the assumptions about producer behavior are different. Because these settings look at different objectives for the producer, they are not directly comparable/there is no baseline to compare to from the related work in Table 2. Previous work also uses regularly softmax as a content-serving rule, so we are in fact using the common baseline from related work, but in an incomparable setting.
>
> We also note that previous work does not have a characterization or even a way to compute Nash equilibria, unlike our work (for example, Hron et al. can only guarantee a weaker notion of equilibrium, which are approximate local equilibria)—therefore, previous work is not directly comparable to our case.  There are other works looking at engagement, but they do not provide techniques to compute an equilibrium or dynamics, rather focusing on other properties of the problem (e.g., Yao et al. characterize the price of anarchy of their game, but do not directly characterizes equilibrium nor how to compute them.) We are happy to update the related work to make this
> clearer.
>
> **Single-Mindedness of Users**
>
> We would like to clarify any potential confusion regarding Assumption 4.1, which introduces single-minded users who focus on a single feature. This assumption is not meant to represent realistic user behavior; instead, it is employed to derive analytically tractable Nash equilibria, providing insights into how producers allocate resources across features and specialize. This assumption is *only* made in section 4, and only to provide intuition as to what the equilibria of the game look like.  It is important to note that all the datasets used in our experiments have users with diverse embeddings. Nevertheless, the equilibria observed under the linear serving rule show that producers distribute their investments across dimensions in proportion to the “weights” assigned to each coordinate, consistent with the findings in Section 4.

---

> > ### Comment · Reviewer_w2au · 2025-01-26
> > **Thank you for the response**
> >
> > The clarifications and additional results provided from the authors have addressed my concerns.
> > I support accepting the paper.

---

> > > ### Author Response · Authors · 2025-01-26
> > >
> > > We deeply thank you for reading our response carefully and for your support!

---

### Decision · Action_Editor_Q2wP · 2025-02-11

**Recommendation:** Accept as is

**Comment:**

All the reviewers found the paper to be a nice contribution and suitable for publication at TMLR.

**Audience:**

It will be of interest to a subset of the TMLR readers.

**Claims And Evidence:**

The paper advances the state of the art in the investigation of content producer incentives in recommender systems, incorporating that the producer reward is based on engagement and that the content-serving rule is continuous.